# How streamflow has changed across Australia since the 1950's: evidence from the network of Hydrologic Reference Stations

Xiaoyong Sophie Zhang[1], Gnanathikkam E. Amirthanathan[1], Mohammed A. Bari[2], Richard M. Laugesen[3], Daehyok Shin[1], David M. Kent[1], Andrew M. MacDonald[1], Margot E. Turner[1], Narendra K. Tuteja[3]

[1]{Environment and Research Division, Bureau of Meteorology, Melbourne, Australia}

[2]{Bureau of Meteorology, Perth, Australia}

[3]{Bureau of Meteorology, Canberra, Australia}

Correspondence to: X. S. Zhang (sophie.zhang@bom.gov.au)

## Abstract

Streamflow variability and trends in Australia were investigated for 222 high quality stream gauging stations having 30 years or more continuous unregulated streamflow records. Trend analysis identified seasonal, inter-annual and decadal variability, long-term monotonic trends, and step changes in streamflow. Trends were determined for annual total flow, baseflow, seasonal flows, daily maximum flow, and three quantiles of daily flow. A distinct pattern of spatial and temporal variation in streamflow was evident across different hydroclimatic regions in Australia. Most of the stations in south-eastern Australia spread across New South Wales and Victoria showed a significant decreasing trend in annual streamflow, while increasing trends were retained within the northern part of the continent. No strong evidence of significant trend was observed for stations in the central region of Australia and northern Queensland. The findings from step change analysis demonstrated evidence of changes in hydrologic responses consistent with observed changes in climate over the past decades. For example, in the Murray-Darling Basin 51 out of 75 stations were identified with step changes of significant reduction in annual streamflow during the middle to late 1990s, when relatively dry years were recorded across the area. Overall, the Hydrologic Reference Stations (HRS) serve as critically important gauges for streamflow monitoring and changes in long-term water availability inferred from observed datasets. A wealth of freely downloadable

hydrologic data is provided at the HRS web portal including annual, seasonal, monthly and
daily streamflow data, as well as trend analysis products, and relevant site information.
**Keywords:** Hydrologic Reference Stations, streamflow variability, trends, step change,
climate change, unregulated catchments, Australia

## 1 Introduction

Assessing changes and trends in streamflow observations can provide vital information for
sustainable water resource management. The influence of diverse environmental factors and
anthropogenic changes on hydrological behaviour makes the investigation into streamflow
changes a challenging task. Trend detection is further complicated from intra-annual, inter-
annual, decadal and inter-decadal variability in streamflow as well as from various
influencing factors that can hardly been analysed separately (WWAP, 2012; Hennessy et al.,

44  2007).

Extensive studies have been undertaken in different parts of the world to analyse long-term
hydrologic trends, and to investigate the possible effect of long-term climate variability on
hydrologic response (Stahl et al., 2010; Birsan et al., 2005; Lins and Slack, 2005; Milly et al.,
2005; Burn and Elnur, 2002). Previous works on streamflow trends draw largely on national
and continental analyses, especially for Europe and North America. Studies of streamflow
variability include analysing trends across Europe (Stahl et al., 2010; Stahl et al., 2012), and
at the national level. For example, Bormann et al. (2011) and Petrow and Merz (2009)
analysed trends under flooding conditions on German rivers. Extensive literature on
hydrological trends have been reported for the UK: Hannaford and Buys (2012) demonstrated
variability in seasonal flow regimes; Hannaford and Marsh (2006, 2008) analysed flow
indicators at an annual resolution, and other studies focused on particular regions (Biggs and
Atkinson, 2011; MacDonald et al., 2010; Dixon et al., 2006; Jones et al., 2006). A wide range
of research on streamflow trends has been published in the USA (Kumar et al., 2009;
Novotny and Stefan, 2007; McCabe and Wolock, 2002) and Canada (Bawden et al., 2014;
Monk et al., 2011; Burn and Hag Elnur, 2002).
Few studies have been published for Australia to-date partly due to limited information on
data records, researches and documentation that could cover all flow regimes. Rivers in some
regions have received close attention only recently. Australia is the driest inhabited continent
with an average annual precipitation of 450 mm and the lowest river flow compared with
other continents (Poff et al., 2006). Water is relatively scarce and is therefore a valuable
resource across the country. Australian streams are characterized by low runoff, high inter-
annual flow variability, and large magnitudes of variations between the maximum and
minimum flows (Puckridge et al., 1998; Finlayson and McMahon, 1988). The wide variety of
unique topographic features combined with variable climates and frequency in weather
extremes result in diverse flow regimes. The recent rise in average temperature and the risk of
future climate variability (BOM, 2015; IPCC, 2014; Cleugh et al., 2011) have added new
dimensions to the challenges already facing communities. Climate variability and its impact
on the hydrologic cycle have necessitated a growing need in Australia to seek evidence of any
emerging trends in river flows.
Chiew and McMahon (1993) examined the annual streamflow series of 30 unregulated
Australian rivers to detect trends or changes in the means. They found that identified changes
in the tested dataset were directly related to the inter-annual variability rather than changes in
climate. The analysis of trends in Australian flood data by Ishak (2010) indicated that about
30% of the selected 491 stations show trends in annual maximum flood series data, with a
downward trend in the southern part of Australia and an upward trend in the northern part.
Several other studies investigated trends of selected streamflow statistics in a particular
region, e.g. southwest Australia (Petrone et al., 2010; Durrant and Byleveld, 2009), southeast
Australia or Victoria (Tran and Ng, 2009; Stewardson and Chiew, 2009). All these studies
addressed the trend analysis of Australian rivers with a limited spatial or temporal coverage of
flow data. A gap in the research remains mainly due to constraints in the access to a dataset of
catchments that can be large enough to represent the diversity of flow regimes across
Australia. Such a dataset would enable a comprehensive and systematic appraisal of changes
and trends in observed river flow records.
The Australian national network of Hydrologic Reference Stations (HRS) was developed by
the Bureau of Meteorology to address this major gap and to provide comprehensive analysis
of long-term trends in water availability across the country (Zhang et al., 2014; Turner et al.,
2012). The HRS website is a one-stop portal to access high-quality streamflow information
for 222 well-maintained river gauges in near-natural catchments. An intention is that the
stations will serve as critically important gauges that record and detect changes in hydrologic
responses to long-term climate variability and other factors.
This paper presents a statistical analysis to detect changes or emerging trends across a range
of flow indicators, based on the daily flow data of 222 sites from the HRS network. The
objective of this study is to provide a nationwide assessment of the long-term trends in
observed streamflow data. Evaluation of past streamflow records and documenting recent
trends will be of benefit in anticipating potential changes in water availability and flood risks.
It is hoped that the findings from trend analysis presented in this paper will inform decision
makers on long-term water availability across different hydroclimatic regions, and be used for
water security planning within a risk assessment framework.

**2 Site selection, data and methods**
**2.1 Hydrologic Reference Stations and data**
The 222 Hydrologic Reference Stations (HRS) were selected from a preliminary list of
potential streamflow stations across Australia according to the HRS selection guideline (SKM
2010). These guidelines specified four criteria for identifying the high quality reference
stations, namely unregulated catchments with minimal land use change, a long period of
record (greater than 30 years) of high quality streamflow observations, spatial
representativeness of all hydro-climate regions, and the importance of site as assessed by
stakeholders. Catchments with extensive basin water use or groundwater pumping were
filtered and not included in HRS catchments, based on the local knowledge of the basin,
stakeholder consultation and land use change analysis. The station selection guidelines were
then      applied      in      four      phases      to      finalise      the      station      list
(www.bom.gov.au/water/hrs/guidelines.shtml). The HRS network will be reviewed and
updated every two years to ensure that the high quality of the streamflow reference stations is
maintained.
Two features were considered in order to define the hydroclimatic regions in HRS: climatic
zones and Australia's drainage divisions. The climatic zones were defined according to
climate classification of Australia based on a modified Koeppen classification system (Stern
et al., 2000). Australia has a wide range of climate zones, from the tropical regions of the
north, through the arid expanses of the interior, to the temperate regions of the south (ABS,
2012). The Australian Hydrological Geospatial Fabric (Geofabric) Surface Catchments
(BOM, 2012) were used to delineate 12 topographically defined drainage divisions
approximating the drainage basins from the Geoscience Australia (2004) definition. The
selection of HRS stations aimed to maximise the geographical extent of the available records.
As shown in Figure 1, the final set of 222 hydrologic reference stations cover all climatic
zones, jurisdictions and most drainage divisions. Since most Australian rivers are located near
the coast, there is a high density of stations along the coast and sparsely distributed stations
across inland areas. One third of the HRS sites are in temperate climate zone, and the majority
of the rest are either in Tropical or Subtropical regions; only a few are located in other climate
zones. The distribution of Hydrologic Reference Stations across multiple hydroclimatic
regions provides data for a comprehensive investigation of long-term streamflow variability
across Australia.
All data used in this study were daily streamflow series of 222 gauging stations from the HRS
network. Table 1 lists the twelve drainage divisions and the number of stations in each
division. The drainage division names are marked on Figure 1. One third of the HRS stations
are located within the Murray-Darling basin, half of the rest are distributed along the east
coast. This is the best compiled long-term quality controlled data for Australia and the trends
derived from this dataset constitute the first such statement on long-term water availability
across Australia.
The earliest record included in the data set is from 1950. Data prior to this has been excluded
due to the common existence of large gaps in the pre-1950 period. All stations included in the
HRS had a target of 5% or less missing data to meet the completeness criteria for high quality
streamflow records. A few stations were included with more than 5% missing data where they
excelled in other criteria such as stakeholder importance or spatial coverage. The periods of
data gaps were filled using a lumped rainfall-runoff model GR4J (Perrin et al., 2003). The gap
filling was found to perform well at most sites. The mean Nash-Sutcliffe coefficient of the
gap-filled time series, when compared to the available original time series data, was 0.72 with
standard deviation of 0.12. The model was calibrated and forced with catchment average
rainfall and potential evapotranspiration from the Australian Water Availability Project
(AWAP) (Raupach et al., 2009).
The study examined all sites using the full length of observations after 1950. Prior to 1950 the
gauge network is generally too sparse for reliable analysis, and analysis periods starting after
mid-1970s are considered too short to calculate meaningful trend values. Although, the data
length of every station was not exactly the same over the continent, but for the stations within
the same region, the data lengths were in more consistent time periods. Data for most of
stations (86%) have very similar time periods. These allow for comparisons on a fairly
consistent basis.
The gap-filled daily flow data were aggregated into annual series based on a water year
calculation. The start month of the water year was defined as the month with the lowest
monthly flow across the available data period. The start month of water year for each region
was recorded in Table 1. The data used in this study were up to end of 2014, so the last water
year cycle ended in 2014. In order to ensure the statistical validity of the trend analysis, all
stations had minimum 30 years of record, with mean time-series length of 48 years and
median time-series length of 46.6 years. The longest record length was 64 years. 25% of the
stations have 50 or more years of record, and 86% stations longer than 40 years data.
Catchment sizes ranged from 4.5 to 232,846 $km^2$ with a median size of 328.6 $km^2$. The
majority (80%) of the stations had an upstream drainage area less than 800 $km^2$, and only
three stations had a drainage area larger than 50,000 $km^2$.
The primary water data has been collected across Australia by many organisations, utilities
and regulators in different states and territories, often to meet the requirements of their own
documented procedures and sometimes with reference to Australian or international standards
or guidelines. The Bureau's role as the national water information provider, has been working
collaboratively with the water industry to develop and promote water information standards
and guidelines to collate, interpret and access nationally consistent data. All data included in
the HRS database are compiled, quality-checked by the Bureau, and therefore are consistent
nationally and over the time. Bureau has developed a set of standard data quality code and
references guides on how it relates to different agencies quality code. The data and the long
term series gathered in this study are the best compiled and quality assured data for HRS
catchments. The analysis and trends derived from the HRS datasets constitute the first
statement on long-term water availability across Australia.

## 184    2.2 Streamflow variables for trend analysis

Long-term climate variability can be reflected through trends in streamflow variables. To
understand the importance of the components of the hydrologic regimes and their potential
link to long-term climate variability, ten streamflow variables were chosen for statistical and
trend analysis. Two variables related to fluctuation of annual flows were annual total flow
($Q_T$) and annual baseflow ($Q_{BF}$). Baseflow was separated from daily total streamflow using a
digital filter based on theory developed by Lyne and Hollick (1979) and applied by Nathan
and McMahon (1990).
Daily streamflow data were analysed to form a group of indicators of daily flow trends. They
were daily maximum flow ($Q_{Max}$), the $90^{th}$ percentile (non-exceedance probability) daily flow
($Q_{90}$), the $50^{th}$ percentile daily flow ($Q_{50}$), and the $10^{th}$ percentile daily flow of each year ($Q_{10}$).
The median daily flow $Q_{50}$ was used in the study instead of daily mean flow because the flow
distribution is skewed and outliers are present.
Four seasonal flow indicators were analysed to examine the seasonal trend patterns. These
variables included summer flow $Q_{DJF}$ (December to February), autumn flow $Q_{MAM}$ (March to
May), winter flow $Q_{JJA}$ (June to August), and spring flow $Q_{SON}$ (September to November).
The trend analysis was applied to the ten hydrologic indicators of streamflow data at each
HRS station.

## 2.3 Trend and data statistical analysis

Changes in streamflow data can occur gradually or abruptly. Statistical significance testing is
commonly used to assess the changes in hydrological datasets (Helsel and Hirsch, 2002;
Monk et al., 2011; Hannaford and Buys, 2012). The Mann-Kendall (MK) trend test (Mann,
1945; Kendall, 1975) was adopted in this study to identify statistically significant monotonic
increasing or decreasing trends (Petrone et al., 2010; Zhang et al., 2010; Miller and Piechota,
2008). In order to ensure the assumption of independence was met for the MK test, the non-
parametric Median Crossing and Rank Difference tests (Kundzewicz and Robson, 2000) were
applied to entire datasets. Both randomness tests consider the long-term persistence as well.
When either of the randomness tests indicated that the time series was not from a random
process, the site was excluded from the MK trend assessment. As this study attempted to
examine patterns in raw historical streamflow records, no further adjustments were made to
account for the non-random structure of data.
The non-parametric MK trend test was used to detect the direction and significance of the
monotonic trend, and the trend line by the non-parametric Sen Slope (Sen, 1968; Theil, 1950)
was used to approximately represent the magnitude of the trend. The trend magnitude was
standardised using the ratio of Sen Slope coefficient to average annual flow in order to make
the change comparable across stations for reporting purposes.
All data were subject to step change analysis to detect any abrupt changes during the record
period. The distribution free CUSUM test (Chiew and Siriwardena, 2005) was applied to
identify the year of change in streamflow series. The significant difference between the
median of the streamflow series before and after the year of change was tested by Rank-Sum
method (Zhang et al., 2010; Miller and Piechota, 2008; Chiew and Siriwardena, 2005). More
information and equations of the statistical tests used in this study can be found in Appendix
A.
In addition to the trend analysis for the ten flow indicators, other statistical data analyses were
included in the HRS web portal to gain a broad understanding of hydrologic regimes.
Aggregated monthly and seasonal flow data were investigated for changes in flow patterns in
different basins or regions. Daily event frequency analyses were used to examine the
variations in daily streamflow magnitude, and daily flow duration curves were presented to
examine changes in daily flow among decades.

## 3 Development of the HRS web portal

A web portal has been developed to house the network of Hydrologic Reference Stations and
provide access to streamflow data, results of analysis, and associated site information. Figure
2 summarises the development process of the HRS network and website. Through a data
quality assurance process following the guidelines and stakeholder consultations, the final list
of 222 streamflow gauging stations was established. A suite of software tools, "the HRS
toolkit" was developed to undertake data aggregation, analysis, trend testing, visualisation and
manipulation. The toolkit is capable of automatically converting the flow variables to
monthly, seasonal and annual totals, and quantifying the step and/or linear changes in the
selected streamflow variables. The toolkit also generated and processed graphical products,
data, statistical summary tables and statistical metadata included in the web portal.
A snapshot of the HRS web portal is shown in Figure 3. The main page was designed with
three parts. A series of links on the top provide the project information. Below this is the
station selector, which facilitates searching for the site of interest by location. The third part is
the product selector containing the core information sections of the website. Several tabs are

offered for users to explore the web portal dependent on their needs and the level of information they require. The daily streamflow data, graphical products, statistics and trend analysis results are available for users to view and download. Information provided on the HRS web portal will assist in detecting long-term streamflow variability and changes at the 222 sites, and therefore supports water planning and decision-making. More information can be found at the website http://www.bom.gov.au/water/hrs.

This web portal provides public access to high quality data and information. It has more than 15,000 graphic products for display. It is carefully designed for the public to have synthesised and easily understandable information on water availability trends across Australia. In order to ensure currency of this web site, streamflow data are updated and reviewed every two years.

## 4 Results

The study to detect long-term streamflow trends was performed on the 222 gauging stations included in the HRS network. This section presents an overview bar-plot of the Mann-Kendall test results for the selected ten hydrologic variables. Maps showing trend detection results and step change analysis for the annual total flow are presented as well as a table listing the stations with significant trends in annual total flow at 1% significance level ($p < 0.01$). In addition, result statistics of trends and step changes were summarised for different regions. Finally, variations in trend among daily flow indicators and seasonal flows were examined.

### 4.1 Overview

A stacked bar-plot is shown in Figure 4 that stratifies the stations by the trend across each streamflow variable. Overall, a consistent pattern is seen across the 10 streamflow variables – the majority of stations have either no trend or a non-random time-series; of the stations with a trend detected, the majority are decreasing.

Patterns of trends were noted in the different flow regimes. Moving through the flow variables from low ($Q_{10}$), median ($Q_{50}$), to high ($Q_{90}$), and onto maximum ($Q_{Max}$), an increasing number of stations were found with no trends, combined with decreasing number for non-random series. The overall number of stations with statistically significant trends was around the same across the median, high, and maximum variables but much lower for the low

flow variable. In the trends of seasonal flows, around one third of stations showed a decreasing trend in spring and a quarter of stations in summer and winter. A significant proportion of stations do show a decreasing trend across the four seasons. Summer flow at a large number of stations showed no trend and three stations with an increasing trend. At most stations the autumn flow time-series were non-random or had no trend, and only about one tenth stations presented a decreasing trend. Due to non-randomness of streamflow variables, a number of stations are not amenable to trend analysis.

## 4.2 Spatial distribution of trends in annual total streamflow

Many hydrological time series exhibit trending behaviour or non-stationarity (Wang, 2006). In fact, trend or step change is one type of non-stationarity (Bayazit, 2015; Rao et al., 2012; Kundzewicz and Robson, 2000). The purpose of trend test in the present study is to determine if the values of a series have a general increase or decrease in the observation time period. Detecting the trends in a hydrologic time series may help us to understand the possible links among hydrological processes, anthropogenic influences and global environment changes. Many of the streamflow time series in this dataset exhibit trends or step-changes in the mean or median.  Abrupt changes and trends in the hydrologic time series could be indicators of hydrologic non-stationarity or long-term gradual changes in the rainfall-runoff transformation processes.

## 4.2.1 Linear trend

Maps were generated showing the trend results for each variable across Australia. As mentioned before, the rank-based non-parametric Mann-Kendall test was used to assess the significance of monotonic trend in the selected flow variables. The magnitude of trend was calculated from Sen Slope. The Rank-Sum test was used to identify the presence of a step change in median of two periods, with the distribution free CUSUM method providing the year of change. Values are reported for sites with Mann-Kendall or Rank-Sum test at higher than 0.1 significant levels for statistically significant monotonic trend or step change. The trend analysis map of annual total streamflow ($Q_T$) displays the direction and significance of a trend (Figure 5) at different levels of significance: $p < 0.01$, $p < 0.05$ and $p < 0.1$. Although trends in $Q_T$ vary across different hydro-climatic regions of the continent, a clear spatial pattern is evident from the map: all stations showing decreasing trends (35% of stations) are in the southern part of Australia and all stations showing increasing trends (4% of stations) in

the northern part, while there is no significant trend visible in the central region of Australia.
The general downward trends observed in southern Australia may have been affected by the
dry period in the last decade in the south-eastern and south-western regions. Stations in the
Murray-Darling Basin demonstrated the strongest decreasing trends with 30 stations
exhibiting high levels of significance at $p < 0.05$.
A set of 22 gauging stations were identified with trends in annual total streamflow at 0.01
significance levels (Table 2). All sites showed consistent direction of change using MK test
and Sen Slope. None of those 22 gauges showed increasing trend. Trends in annual baseflow
were found to be similar to the results of annual totals when a significant trend was detected.
Baseflow index was listed in Table 2 calculated by the ratio of baseflow to total flow, and the
trend results of baseflow was indicated at the top right corner. The number of stations
showing significant declining trends in baseflow conditions was less than it was for annual
total flow. However, some time-series of annual baseflow were non-random and therefore not
available for further trend testing.

### 4.2.2 Step change

Step change analysis was applied to all sites where the time series data was random to give
comparable results of gradual and abrupt changes in annual total flows. The Rank-Sum test
was used to identify the presence of a step change in the median of two periods, with the
distribution free CUSUM method providing the year of change. Values were reported for sites
with Rank-Sum test at 0.1 significance levels or higher. Figure 6 shows the results of step
change analysis, where colours indicate the year of change appearing in each decade, and
upward arrows represent increased median values after the year of change and vice versa.
The step change map reveals a definite spatial pattern in the location of stations that exhibited
a significant step change. As expected, the direction and significance of step-changes is
consistent with the Mann-Kendall results for most stations. The identified years of step
changes appear to show spatial groupings at different divisions. Table 2 gives the Rank-Sum
test (RS) results and lists the year of change for the 22 stations. The majority of stations in
southeast Australia were characterised with step changes in mid-1990s, when the so-called
"millennium drought" (BOM and CSIRO, 2014; SEACI, 2011) started to dominate the
weather in this region. It has been reflected in Table 2: 13 of 22 stations presented the years of
the step change in 1996, which was clearly the most dominating year. In Ummenhofer et al.
(2009) where the most severe drought in Australia was discussed, the affected region referred
to as south-eastern Australia is defined as the land region enclosed within $35° - 40°S$ and
$140° - 148°E$. Stations inside that defined region exhibited the major feature of a step change
in the 1990s which can be seen by the purple downward arrows dominating Figure 6, stations
outside the region exhibited step changes with mixed years of changes. This included a good
number of 1970s changes at the northeast New South Wales, 1980s changes at the south east
coast of Queensland, and 2000s changes in South Australia. Five stations in south-west West
Australia had a key feature of 1975 step change, which might be partly due to the observed
rainfall decline since the mid-1970s. It was also noted that most stations located in the
Northern Territory and some in the northeast coast of Queensland showed a significant
increasing step change.
Figure 7 summarizes the results of the trend test on the flow variable of annual total
streamflow. It describes the percentage and number of stations with an upward or downward
trend or step change in each region. The Australian states on the x axis were organised from
left to right in the order of the increasing number of stations in each state. Across all the eight
regions investigated in this study, the stations located in southern part of the country
displayed a decreasing trend and step change persistently. These regions included Australian
Capital Territory (ACT), South Australia (SA), Tasmania (TAS), southwest of Western
Australia (WA), New South Wales (NSW), and Victoria (VIC). The number of stations with
significant downward step changes was similar to, or slightly higher than the ones with
detected trends. Upward changes were only observed at the north part of continent: most
stations in Northern Territory (NT), one station with weak trend at north WA and one at north
Queensland (QLD). Mixed patterns of upward and downward step changes were detected in
Queensland, which has the most diverse climatic conditions.
**4.3 Spatial distribution of trends in daily flows and seasonal flows**
Trend analysis maps shown in Figure 8 decompose trends of daily flow for $Q_{Max}$, $Q_{90}$, $Q_{50}$ and
$Q_{10}$. In general, the identified trends were spatially consistent with the trend pattern in $Q_T$:
with upward trends in the north-west and downward trends in the south-east, south-west and
Tasmania. The $Q_{50}$ and $Q_{10}$ series were notable for the number stations with  non-random
time-series and therefore an invalid MK test result, this could be seen most dramatically in
Figure 8d, and was due to the higher correlation of the time-series. This daily flow trend
analysis indicated similar results to previous studies (Tran and Ng, 2009; Durrant and
Byleveld, 2009) for the respective sites and flow statistics.
The analysis of maximum daily flow $Q_{Max}$ could be considered as analysis of extreme flow as
this series contains the maximum value for each year. The general pattern of trends in $Q_{Max}$
was in accordance with the preliminary trend analysis results in Ishak (2010), which
suggested that about 30% of selected stations showed trend in $Q_{Max}$, with downward trend in
the southern part of Australia and upward trends in the northern part (Figure 8a).
The spatial distribution of trends in seasonal flows was investigated to disaggregate total flow
series into seasons (Figure 9). The broad pattern from the analysis was a collection of few
upward trends in the north and predominant downward trends generally in the south. Across
the four seasonal variables, spring flow ($Q_{SON}$), winter flow ($Q_{JJA}$), summer flow ($Q_{DJF}$), and
autumn flow ($Q_{MAM}$) were in the sequence of decreasing number of detected downward
trends. All seasons presented significant downward trends mostly in the southern parts of
Australia, with autumn having fewer than others.

**5. Discussion**
We have demonstrated a comprehensive statistical and trend analysis in long-term streamflow
data for 222 unregulated river gauges from the HRS national network. Ten streamflow
variables were examined to detect underlying changes or trend in streamflow and to identify
spatial variations across Australia. Evidence from previous research and this current study
raises an important question: what could be the key driver of the detected changes in
Australian streamflow data? Though it is beyond scope of this study to examine underlying
mechanisms linking flow, climate and other factors, some remarks may help to provide
valuable information for understanding and interpreting Australian hydrology.
**5.1 Evidence for trends in hydrological records Australia**
Numerous studies have analysed Australian streamflow data to detect any existing trends in
hydrologic records. Chiew and McMahon (1993) examined trends in annual streamflow of 30
sites across Australia and no clear evidence of changes were suggested with the data available
at that time. Haddad et al. (2008) reported a decreasing trend in many Victorian stations of
annual maximum floods particularly after 1990. Tran and Ng (2009) also showed a
consistently decreasing trend among 9 streamflow statistics of 14 stations in a Victorian
region, but indicated the result was not able to relate the effect of global climate change with
the decreases in streamflow. Durrant and Byleveld (2009) analysed post-1975 flow record at
29 sites across south-west Western Australia; they indicated the majority of sites show a
consistent regional reduction in streamflow. Silberstein et al. (2012) further computed
simulations of runoff from 13 major river basins in south-western Australia. They found that
the reduction in runoff for the study region is likely to continue under projected future
climates. Pui et al. (2011) detected changes in annual maximum flood data of 128 stations in
NSW according to multiple climate drivers. Ishak et al. (2010, 2013) presented trend analysis
in annual maxima flood series data from 491 stations in Australia, and suggested much of the
observed trend may be associated with the climate modes on annual or decadal timescales.
Commonality and differences were found from this study when compared with previous
streamflow trend studies across Australia. This could be expected given the different selection
of flow statistics, gauge location, data length, employed techniques and methodology. For
example, to examine the trends in south-west Western Australia (SWWA), Durrant and
Byleveld (2009) has investigated 29 sites in the area using post-1975 data, whilst this paper
considered the full record of data since 1950 and the full water year was used. Owing to the
different data record periods used in trend analysis, seven stations in Durrant and Byleveld
(2009) showed a possible increase, while in this study a homogenous spatial pattern of
downward trends was revealed across the SWWA. Three stations in common were examined
in both studies. The streamflow data of Yarragil Brook at Yarragil Formation, AWRC ID
614044 (Australian Water Resources Council), in Murray River basin was a non-random
series, which was strongly biased by the 1975 step change. When only looking at the runoff of
post-1975 period at this site, it revealed a very weak decreasing trend, which was similar to
the result of Durrant and Byleveld (2009). Carey Brook at Staircase Road (608002) in
Donnelly River basin had similar time series data starting from the mid-1970s in both studies.
A slight decreasing linear trend and a 1997 step change at 0.05 significance level was
identified in this study. No statistically significant trend was detected in Durrant and Byleveld
(2009), which could be attributed to the limited record until 2008 and not considering the
recent years of 2010, 2011 and 2012 that were relatively dry. The results were in agreement in
both studies showing no strong decreasing trend for the Kent River at Styx Junction (604053).
At this site the 1975 change was not predominant.
The results of this study have demonstrated the main characterisation of hydrological change
of river flows across Australia since the 1950's. Overall, most of the downward trends in $Q_T$
appeared within or very close to the temperate climate zone, while upward trends were in the
tropical region. And a large number of step changes occurred in 1996 across southeast
Australia.
**5.2 Further remarks on detected trends**
Many factors could contribute to changes in runoff characteristics, ignoring climate change as
well as low-frequency climate variability and human intervention in river basins compromise
the assumption of stationarity (Ajami et al., 2016; Bayazit, 2015; Smetterm et al., 2013;
Ummenhofer et al., 2009). Higher temperature and changes in precipitation or other climate
variables impact on the rainfall-runoff process directly, and indirectly causing changes in
flora, relief and soil erosion. Changes in catchment characteristics, either naturally or under
human influence such as farm dams, can also have an important influence on water flow.
Moreover, high climate variability and recent climate trends has been observed in Australia,
as the continent is effected by many different weather systems and is driven by many
significant climate features (CSIRO and BOM, 2015; BOM, 2015). Accordingly, hydrologic
data of Australian rivers generally have strong natural variability, subject to data availability
and quality. All these factors make it challenging to detect changes or trends in streamflow
data. Even if a trend is identified, it is difficult to attribute changes to any specific cause, as
global warming and other regional or local changes are contributing to the hydrologic process.
The long-term rainfall trends (1970-2015) in annual total rainfall Australia has been analysed
and published (http://www.bom.gov.au/climate/change/#tabs=Tracker&tracker=trend-maps).
The identified trend patterns in annual total streamflow are spatially consistent with trends in
annual total rainfall, where most of eastern and south-western Australia has experienced
substantial rainfall declines since 1970; while north-western Australia has become wetter over
this period. This similarity implies that hydrological variability is closely related with changes
in rainfall patterns.
The spatial pattern of trends matched the rainfall records maps that indicated rainfall
deficiency in the south in the last decade comparing the historical records (Cleugh et al.,
2011). Similar rainfall changes were also observed as shown in the recent CSIRO sustainable
yield study projects (CSIRO, 2015). Drought conditions, the most persistent rainfall deficit
since the start of the 20[th] century, persisted in the south-east and south-west of the continent
from around 1996 to 2010, which might be attributed to the detected change in streamflow.
This could be the reason that most of the gauging stations in southern Australia and southeast
of Queensland showed a significant decreasing trend in annual streamflow. It was also found
that positive trends observed at many locations in northern Australia could be related to
increased rainfall in this part of Australia during the last decade (SEACI, 2011). Other
changes such as within-year rainfall variation and increase in temperature may have played a
role in affecting the hydrologic cycle.
Whilst it is a possible explanation, it is not explicit that climate change is the only cause of
significant trends in streamflow. There are many other factors that may affect streamflow, for
example, natural catchment changes, climate variability, data artefacts and other influences.
Site specific comparison of rainfall, PE, and temperature may help to improve the
understanding of the underlying causes of trends in hydrological variables. Further
investigation would be required to discover the potential causes of detected trends, which was
beyond the scope of this study.
Under the Water Act (2007), the Australian Bureau of Meteorology has responsibility for
compiling and disseminating comprehensive water information nation-wide. Hydrologic
Reference Stations (HRS) is an initial step to build up the national river data network. The
network of HRS, which the present study was based on, is the first operational website in
Australia as a national river flow data repository. It provides an excellent foundation for water
planning and research – particularly in trend detection and the possibility to link to large scale
atmospheric and climate variables. The information on the HRS website can be used as a test
bed for model development, hydrological non-stationarity assessments and many other
research interests.

**6. Conclusions**
This study investigated the streamflow variability and inferred trends in water availability for
222 gauging stations in Australia with long term and high quality streamflow records. The
results present a systematic analysis of recent hydrological changes in greater spatial and
temporal details than previously published for Australian rivers. Implications of the findings
should aid decision making for water resources management, especially when considering the
results in the context of climate variability.
The main findings of the study are:
● The spatial and temporal trends in observed streamflow varied across different
hydro-climatic regions in Australia (Figure 1 and Figure 5). As a short summary of the

trend test results in annual streamflow ($Q_T$) over the continent, most of the increasing trends were observed in northern part of Northern Territory, while there was only one weak trend visible in the northern region of Western Australia and Queensland. However, in south-eastern Queensland there was a significant decreasing trend. Most of the gauging stations in New South Wales, Victoria, south-east South Australia, south-west Western Australia, and north-west Tasmania showed a significant decreasing trend in annual streamflow. In central Australia, north Queensland and South Tasmania, most of the stations showed no significant trend in annual streamflow.

● The temporal trends also varied between different components of streamflow – annual total, daily maximum ($Q_{Max}$), high, median and low flows ($Q_{90}$, $Q_{50}$, $Q_{10}$), baseflow ($Q_{BF}$) and seasonal flows ($Q_{JJA}$, $Q_{SON}$, $Q_{DJF}$, $Q_{MAM}$). Out of 222 stations, only 7 showed an increasing trend, 90 decreasing and 98 no trend in total annual streamflow. The annual daily maximum streamflow showed decreasing trends at 67 stations while the low flow and baseflow components showed decreasing trends at 18 and 73 stations respectively. Trends also varied between different seasonal flows and also across different hydro-climatic regions. Seasonal flow maps were dominated with decreasing trends. A few stations in northern Australia presented increasing trend for spring, summer and winter flow, while no stations were found with increasing trend for autumn flow ($Q_{MAM}$) anywhere in Australia.

● The analysis of step changes revealed definite regional patterns: The majority of stations in the southern parts of Australia were characterised with downward step changes, while stations with significant upward step changes were seen in the Northern Territory and some of the northeast coast of Queensland.

● The web portal (http://www.bom.gov.au/water/hrs) displays all the graphical products, tables, and statistical test results of all 222 stations. It contains a comprehensive unique set of graphical products for linear trends and step change.

The streamflow trends evident from the statistical data analysis showed some parallels with climate variability patterns that the country experienced through recent decades. Long-term trends in water availability across different hydroclimatic regions of Australia reported in this study are derived purely from observations, not derived from models which can invariably be influenced by biases. The high quality streamflow data of HRS and the results from this

analysis on streamflow variability provide critical information for water security planning and
for prioritising water infrastructure investments across Australia.

**Appendix A: Statistical tests**
**A1. Median Crossing Test**
This method tests for randomness of a time series data. It is a non-parametric test. The n time
series values ($X_1$, $X_2$, $X_3$... $X_n$) are replaced by '0' if $X_i < X_{median}$ and by '1' if $X_i > X_{median}$. If
the time series data come from a random process, then the count 'm', which is the number of
times 0 is followed by 1 or 1 is followed by 0, is approximately normally distributed with:
Mean: $\mu = \dfrac{(n-1)}{2}$
Standard deviation: $\sigma = \dfrac{(n-1)}{4}$
The z-statistic is therefore defined as:
$$z = \dfrac{|(m-\mu)|}{\sigma^{0.5}}.$$
**A2. Rank Difference Test**
This method also tests for randomness of a time series data. It is a non-parametric test. The n
time series values ($X_1$, $X_2$, $X_3$... $X_n$) are replaced by their relative ranks starting from the
lowest to the highest ($R_1$, $R_2$, $R_3$... $R_n$). The statistic 'U' is the sum of the absolute rank
differences between successive ranks:
$$U = \sum_{i=2}^{n} |R_i - R_{i-1}|$$
For large n, U is normally distributed with:
Mean: $\mu = \dfrac{(n+1)(n-1)}{3}$
Standard deviation: $\sigma = \dfrac{(n-2)(n+1)(4n-7)}{90}$
The z-statistic* is therefore defined as:
$$z = \frac{|(U - \mu)|}{\sigma^{0.5}}.$$

## A3. Mann-Kendall Test

This method tests whether there is a trend in the time series. It is a non-parametric rank-based
test. The n time series values $(X_1, X_2, X_3 \ldots X_n)$ are replaced by their relative ranks starting
from the lowest to the highest $(R_1, R_2, R_3 \ldots R_n)$.

560        The test statistic S is defined as:

$$S = \sum_{i=1}^{n-1} [\sum_{j=i+1}^{n} \mathrm{sgn}(R_i - R_j)]$$

| 563 | where | $\mathrm{sgn}(y) = 1$ for $y > 0$ |
| 564 | | $\mathrm{sgn}(y) = 0$ for $y = 0$ |
| 565 | | $\mathrm{sgn}(y) = -1$ for $y < 0$ |
| 566 | | $\mathrm{sgn}()$ is the signum function. |

If there is a trend in the time series (i-e the null hypothesis $H_o$ is true), then S is
approximately normally distributed with:

569        Mean: $\mu = 0$

570        Standard deviation: $\sigma = \dfrac{n(n-1)(2n+5)}{18}$

The z-statistic* is therefore:
$$z = \frac{|S|}{\sigma^{0.5}}$$
A positive value of S indicates that there is an increasing trend and vice versa.

## A4. Distribution Free CUSUM Test

This method tests whether the means in two parts of a record are different for an unknown
time of change. It is a non-parametric test. Given a time series data $(X_1, X_2, X_3 \ldots X_n)$, the test
statistic $V_k$ is defined as:

$$V_k = \sum_{i=1}^{k} \mathrm{sgn}(X_i - X_{median})]$$

| 581 | where | $\mathrm{sgn}(y) = 1$ for $y > 0$ |
| 582 | | $\mathrm{sgn}(y) = 0$ for $y = 0$ |
| 583 | | $\mathrm{sgn}(y) = -1$ for $y < 0$ |
| 584 | | $X_{median}$ is the median value of the $X_i$ data set. |

The time at which 'max$|V_k|$' occurs is considered as the time of change. The distribution of $V_k$
follows the Kolmogorov-Smirnov two-sample statistic (KS = (2/n) max$|V_k|$). A negative value
of $V_k$ indicates that the latter part of the record has a higher mean than the earlier part and vice
versa.
**A5. Rank-Sum Test**
This method tests whether the medians in two different periods are different. It is a non-
parametric test. The time series data is ranked to compute the test statistic. In the case of ties
the average of ranks are used. The statistic S is the sum of ranks of the observations in the
smaller group. The theoretical mean and standard deviation of S under $H_o$ for the entire
sample is given as:
Mean: $\mu = \dfrac{n(N+1)}{2}$
Standard deviation: $\sigma = \left[\dfrac{nm(N+1)}{12}\right]^{0.5}$
where n and m are the number of observations in the smaller and larger groups
respectively. The standardised form of the test statistic, Z* is computed as:
$Z = (S - 0.5 - \mu) / \sigma$      if $S > \mu$
$Z = 0$      if $S = \mu$
$Z = |S + 0.5 - \mu| / \sigma$      if $S < \mu$
Z is approximately normally distributed.
**Acknowledgements**
The primary streamflow data for this study were provided by the national and state water
agencies. The Hydrologic Reference Stations website was developed in consultation with
University of Melbourne, CSIRO Land and Water, Department of the Environment (DOE)
and about 70 other stakeholders. Special thanks go to Emeritus Professor Tom McMahon for
his contribution to the HRS technical review. We also gratefully acknowledge the input from
AMDISS team, Water Data, and Geofabric teams of the Bureau of Meteorology.

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

Table 1: Metadata of the drainage divisions and Hydrologic Reference Stations

| Division map code | Drainage division names | Mean annual rainfall (mm) (1976-2005)* | Mean elevation (m) | Number of HRS stations | Water year start month | Smallest catchment area (km$^2$) | Largest catchment area (km$^2$) |
|---|---|---|---|---|---|---|---|
| I | Northeast Coast | 764 | 173 | 42 | October | 6.6 | 7486.7 |
| II | Southeast Coast | 599 | 323 | 44 | March | 4.5 | 4660.0 |
| III | Tasmanian | 1519 | 199 | 12 | February | 18.3 | 775.3 |
| IV | Murray-Darling | 479 | 260 | 75 | March | 26.3 | 35238.9 |
| V | South Australia Gulf | 344 | 269 | 5 | February | 5.3 | 187.4 |
| VI | Southwest Coast | 329 | 365 | 13 | March | 14.1 | 1786.0 |
| VII | Indian Ocean | 369 | 162 | 0 | (No data) | (No data) | (No data) |
| VIII | Timor Sea | 520 | 339 | 13 | September | 65.4 | 47651.5 |
| IX | Gulf of Carpentaria | 674 | 293 | 13 | October | 170.0 | 43476.2 |
| X | Lake Eyre | 429 | 312 | 5 | October | 434.9 | 232846.3 |
| XI | North Western Plateau | 456 | 359 | 0 | (No data) | (No data) | (No data) |
| XII | South Western Plateau | 321 | 297 | 0 | (No data) | (No data) | (No data) |

* Calculation was based on rainfall data from BOM climate website http://www.bom.gov.au/

Table 2: Results of trend analysis for stations showing MK trend test at 0.01 significance level in annual total streamflow

| Div | State | Basin | Site name | Station ID | Area (km²) | Data series Start year | Data series End year | Ave. annual flow (GL/yr) | BF index | Trend MK | Trend Sen Slp | Step change RS | Step change year |
|---|---|---|---|---|---|---|---|---|---|---|---|---|---|
| II | VIC | Snowy River | Buchan River at Buchan | 222206 | 850 | 1951 | 2014 | 140.0 | 0.45-- | **↓ | -1.2% | ** | 1976 |
| | VIC | Mitchell-Thomson Rivers | Tambo River at Swifts Creek | 223202 | 899 | 1951 | 2014 | 77.1 | 0.46-- | **↓ | -1.3% | ** | 1978 |
| | VIC | Werribee River | Lerderderg River at Sardine Creek O'brien Crossing | 231213 | 152 | 1959 | 2014 | 25.9 | 0.34-- | **↓ | -1.7% | ** | 1996 |
| | VIC | Hopkins River | Mount Emu Creek at Mena Park | 236213 | 448 | 1974 | 2014 | 13.6 | 0.18-- | **↓ | -3.3% | ** | 1996 |
| | VIC | Glenelg River | Jimmy Creek at Jimmy Creek | 238208 | 23 | 1951 | 2014 | 3.4 | 0.47**,↓ | **↓ | -1.5% | ** | 1996 |
| | SA | Millicent Coast | Mosquito Creek at Struan | A2390519 | 1550 | 1971 | 2014 | 21.7 | 0.25-- | **↓ | -2.6% | ** | 1992 |
| | SA | Millicent Coast | Stony Creek at Woakwine Range | A2390523 | 485 | 1973 | 2014 | 4.8 | 0.55**,↓ | **↓ | -2.6% | ** | 1990 |
| III | TAS | Smithton-Burnie Coast | Black River at South Forest | 314213 | 318 | 1968 | 2014 | 194.1 | 0.38-- | **↓ | -1.0% | ** | 1992 |
| IV | NSW | Upper Murray | Maragle Creek at Maragle | 401009 | 217 | 1951 | 2014 | 35.9 | 0.47**,↓ | **↓ | -1.2% | ** | 1996 |
| | VIC | Kiewa River | Flaggy Creek at Myrtleford Road | 402217 | 26 | 1970 | 2010 | 4.0 | 0.42**,↓ | **↓ | -2.6% | ** | 1996 |
| | VIC | Goulburn | Mollison Creek at Pyalong | 405238 | 164 | 1972 | 2014 | 19.5 | 0.29-- | **↓ | -3.8% | ** | 1996 |
| | VIC | Goulburn | Major Creek at Graytown | 405248 | 288 | 1971 | 2014 | 13.2 | 0.10**,↓ | **↓ | -2.9% | ** | 1996 |
| | VIC | Goulburn | Brankeet Creek at Ancona | 405251 | 122 | 1973 | 2014 | 14.8 | 0.45-- | **↓ | -2.4% | ** | 1996 |
| | VIC | Campaspe River | Axe Creek at Longlea | 406214 | 237 | 1972 | 2014 | 13.4 | 0.18-- | **↓ | -4.0% | ** | 1996 |
| | VIC | Loddon River | Creswick Creek at Clunes | 407214 | 300 | 1951 | 2014 | 24.0 | 0.32**,↓ | **↓ | -1.9% | ** | 1996 |
| | VIC | Loddon River | Joyces Creek at Strathlea | 407230 | 156 | 1973 | 2014 | 9.2 | 0.17-- | **↓ | -3.3% | ** | 1996 |
| | NSW | Lachlan | Abercrombie River at Abercrombie | 412028 | 2631 | 1951 | 2014 | 277.0 | 0.30-- | **↓ | -1.6% | ** | 1978 |
| | NSW | Lachlan | Abercrombie River at Hadley | 412066 | 1630 | 1960 | 2014 | 169.8 | 0.29-- | **↓ | -1.7% | ** | 1978 |
| | VIC | Avon | Richarson River at Carrs Plains | 415226 | 125 | 1971 | 2014 | 3.7 | 0.04**,↓ | **↓ | -2.7% | ** | 1996 |
| | VIC | Wimmera | Concongella Creek at Stawell | 415237 | 244 | 1976 | 2014 | 9.1 | 0.12**,↓ | **↓ | -3.8% | ** | 1996 |
| VI | WA | Murray River (WA) | Harvey River at Dingo Road | 613002 | 148 | 1970 | 2014 | 29.7 | 0.58-- | **↓ | -1.6% | ** | 1993 |
| | WA | Swan Coast | Canning River at Glen Eagle | 616065 | 537 | 1953 | 2014 | 18.9 | 0.36**,↓ | **↓ | -1.7% | ** | 1975 |

* Significant at p < 0.05    ** Significant at p < 0.01
- baseflow series non-random    ○ baseflow no trend
↓ decrease trend    ↑ increase trend

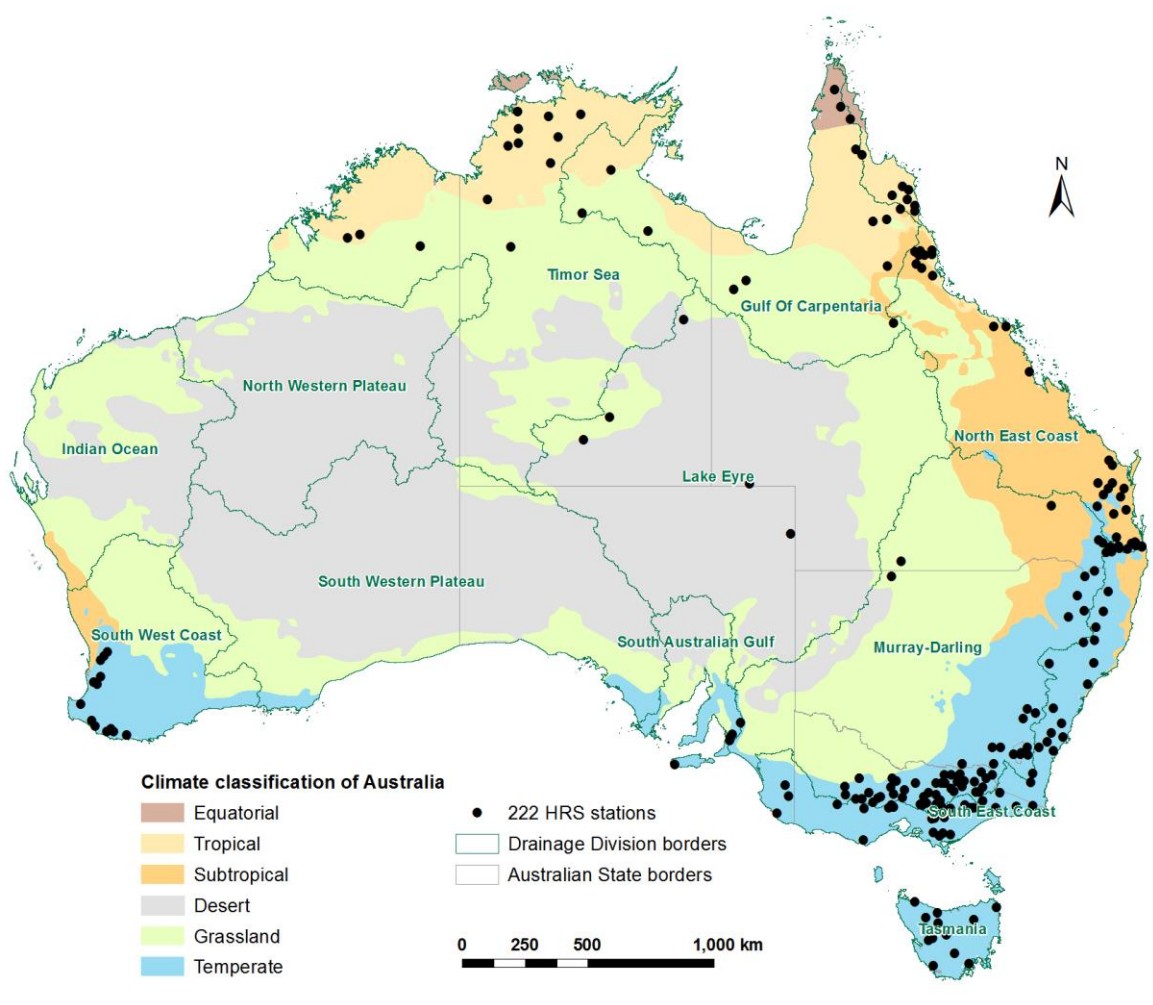



Figure 1: Location of the 222 high quality streamflow reference stations, the climatic regions
and Australia drainage divisions (Geofabric Surface Hydrology Catchments, Geofabric V2.1,
BOM 2012)

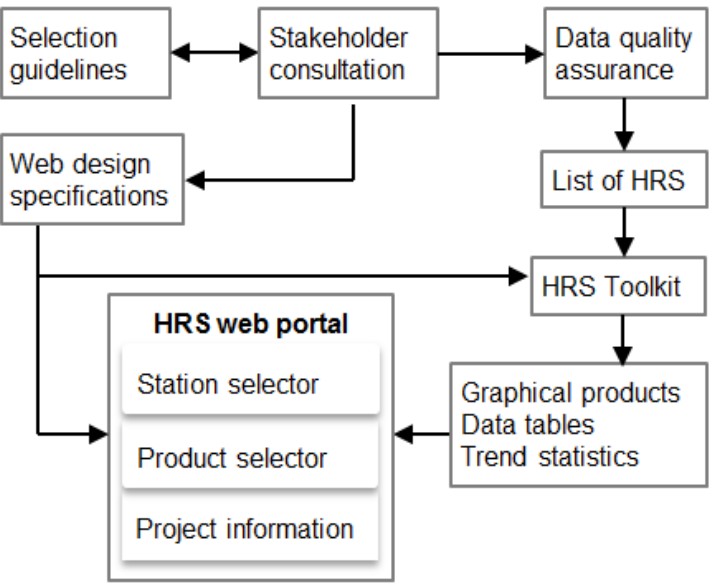



Figure 2: Framework of developing Hydrologic Reference Stations

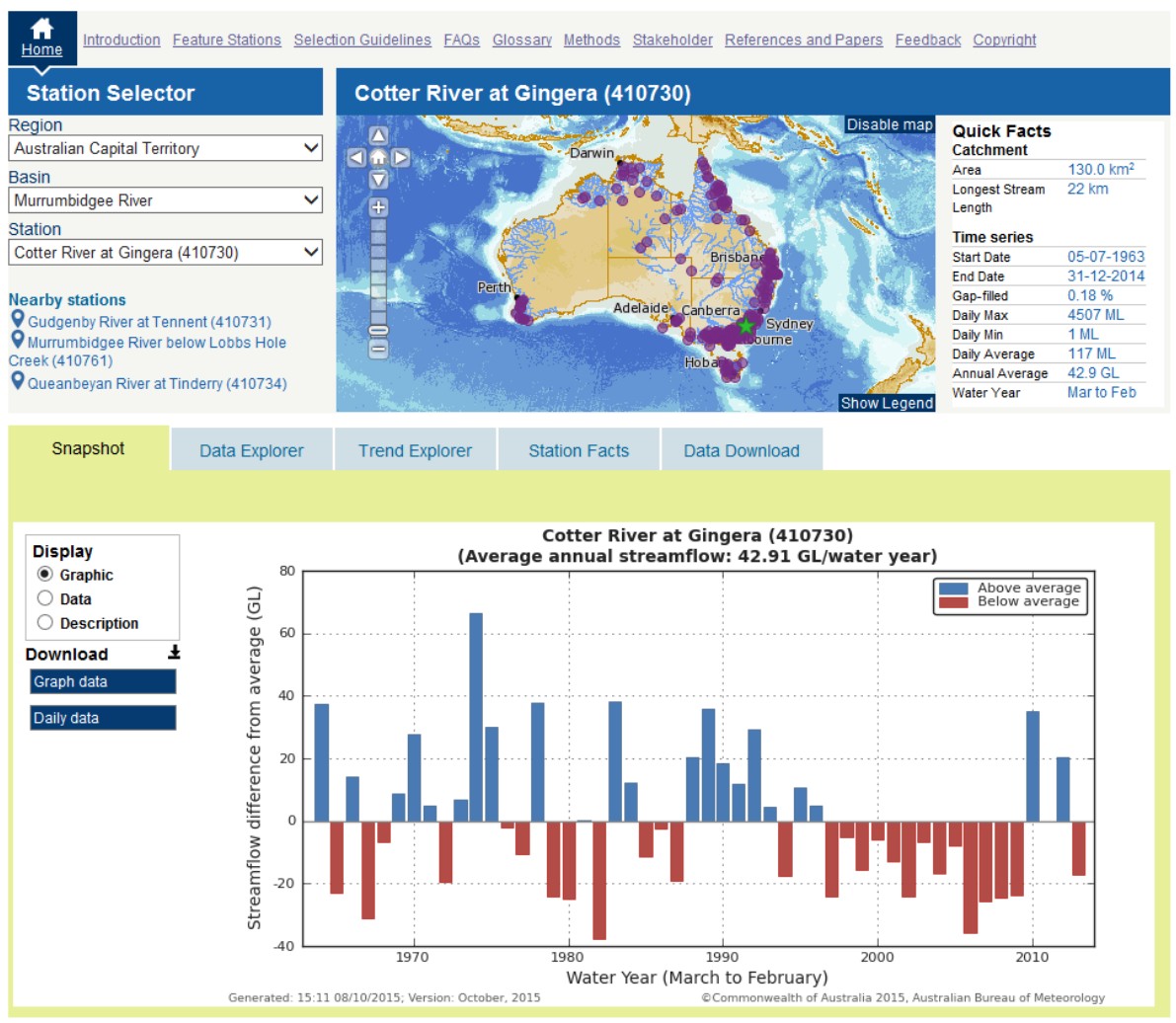



Figure 3: Snapshot of the HRS web portal

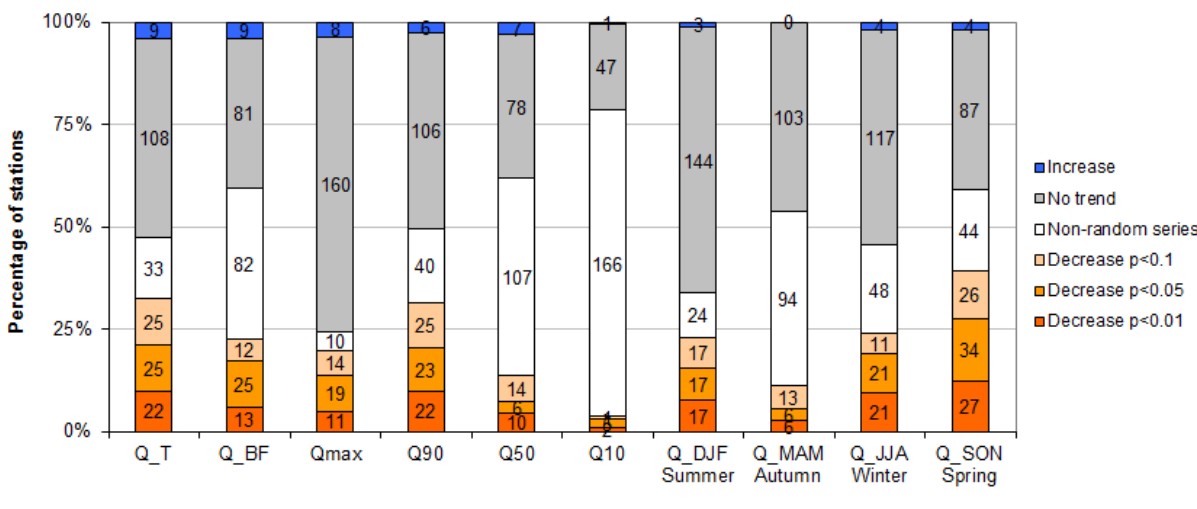



Figure 4: Stacked bar-plot summarizing the MK trend test results for the 222 HRS stations,
with data labels showing the number of stations in each category (Q_T: annual total flow,
Q_BF: annual baseflow, Qmax: daily maximum flow, Q90: 90th percentile daily flow, Q50:
50th percentile daily flow,Q10: 10th percentile daily flow, Q_DJF: summer flow, Q_MAM:
autumn flow, Q_JJA: winter flow, Q_SON: spring flow)

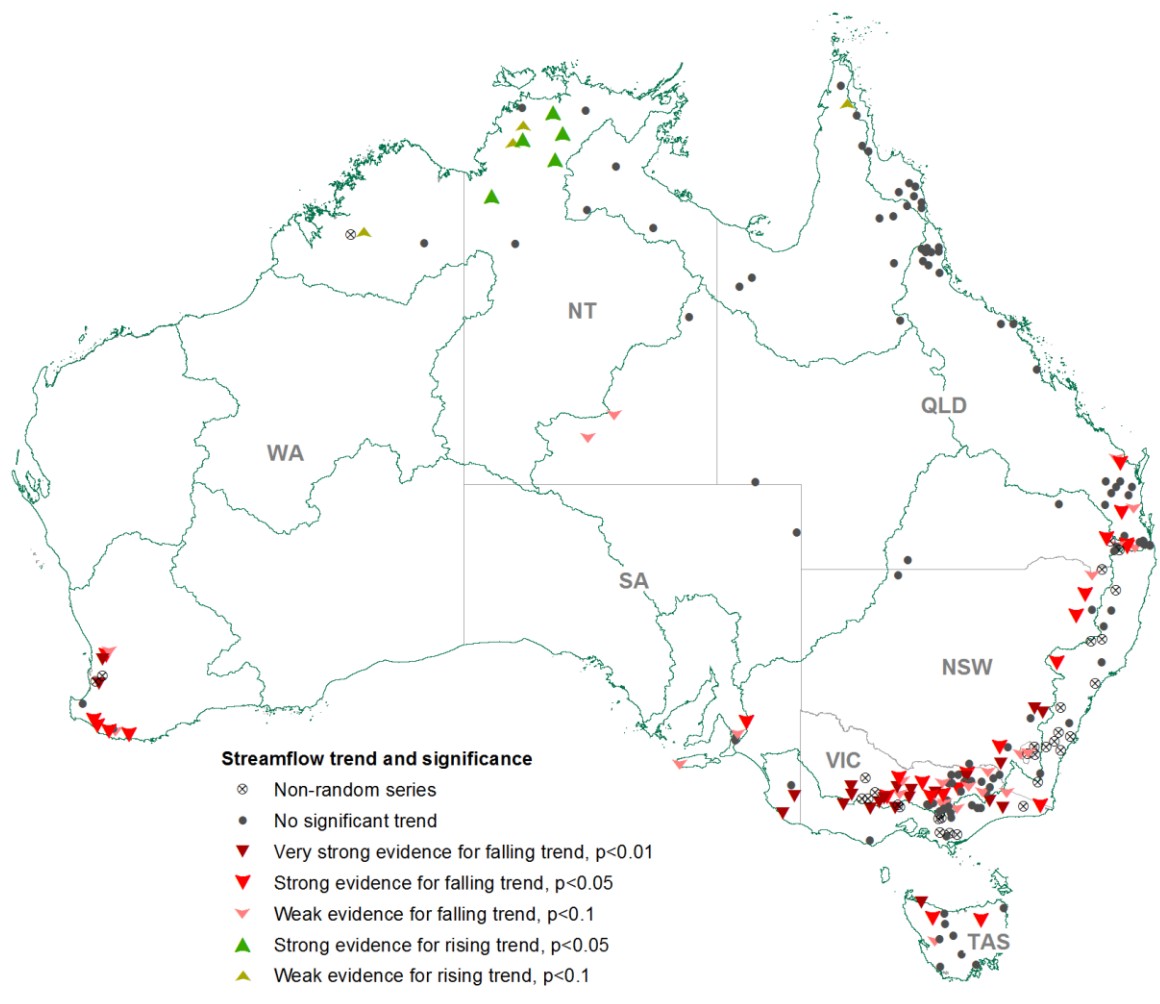



Figure 5: Spatial variation in trend results of annual total flow ($Q_T$), trends were shown in
significance levels at 0.01, 0.05, and 0.1

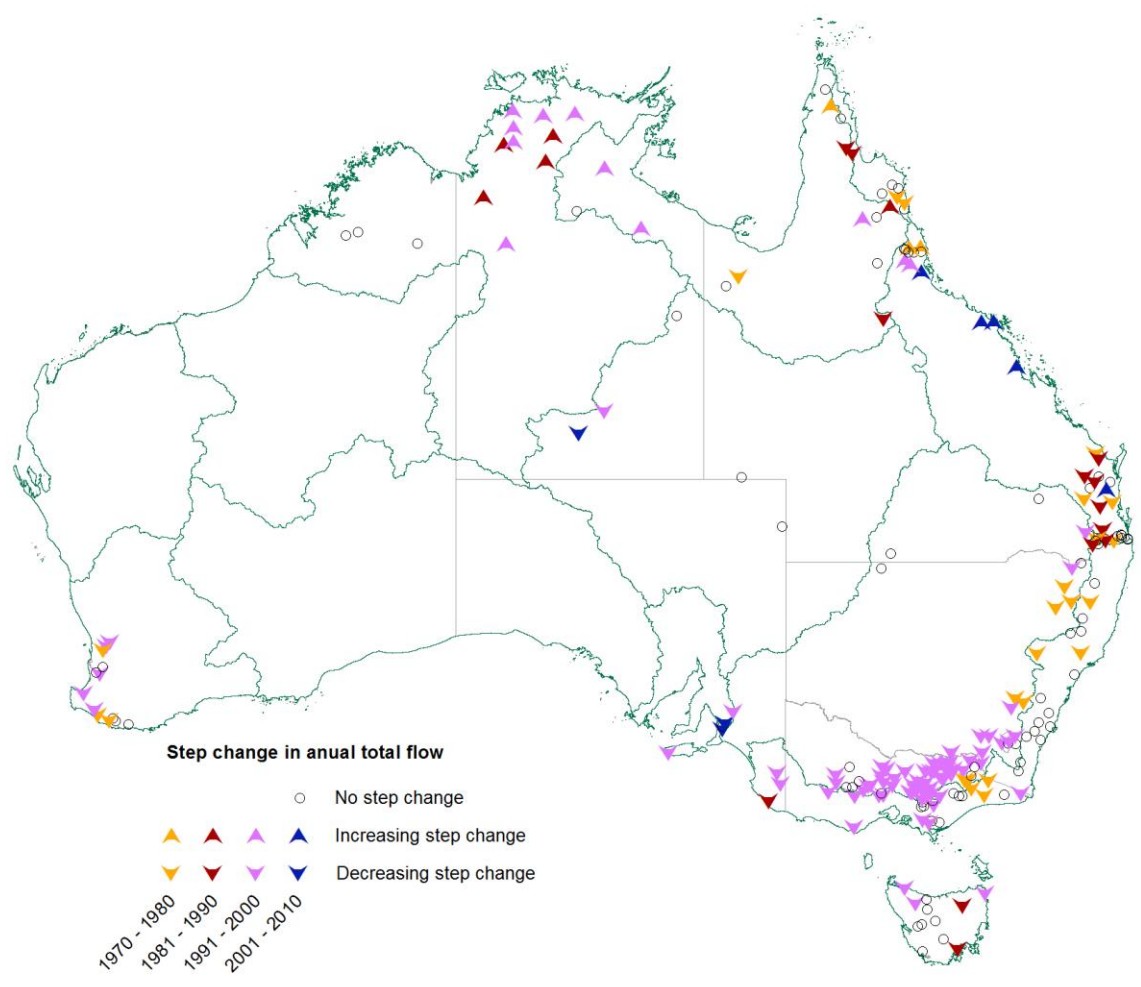



Figure 6: Variations of step change in annual total flow ($Q_T$), with the year of change
indicated in each decade

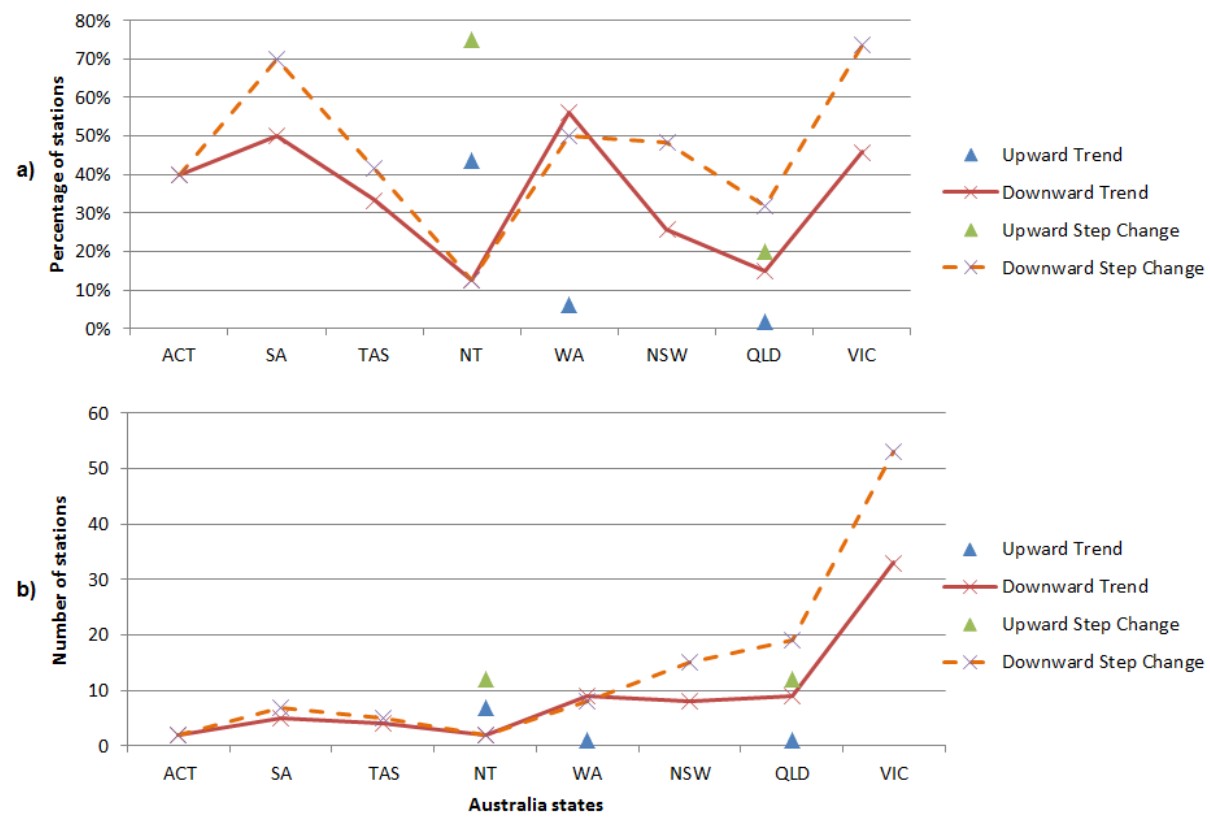



Figure 7: a) Percentage and b) number of stations showing significant upward and downward
trends or step changes in Australia states and territories





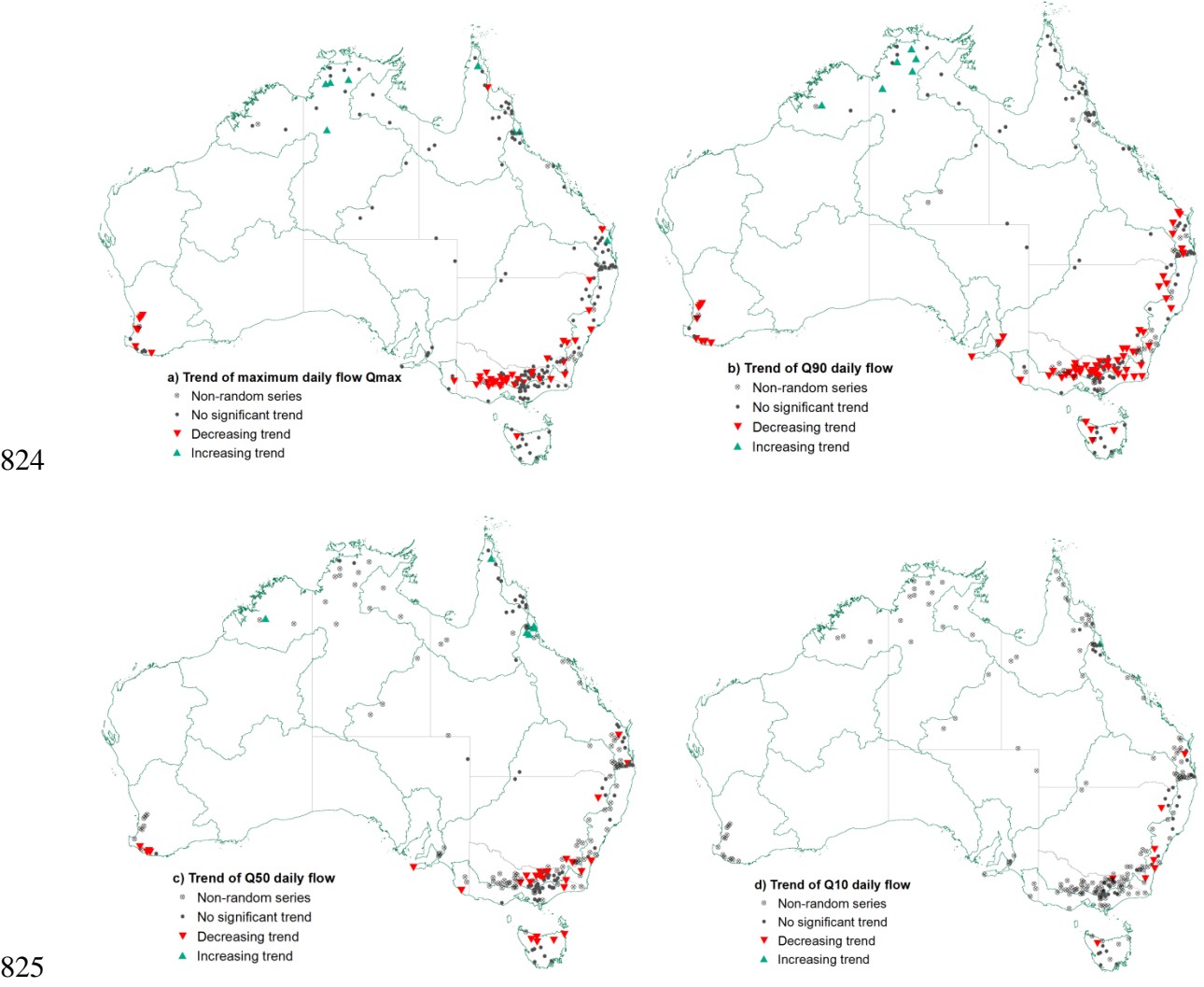




Figure 8: Maps showing trends of daily flow in various magnitude categories a) maximum daily flow $Q_{Max}$; b) $Q_{90}$ daily flow; c) $Q_{50}$ daily flow; d) $Q_{10}$ daily flow at 10% significant level (p<0.1)










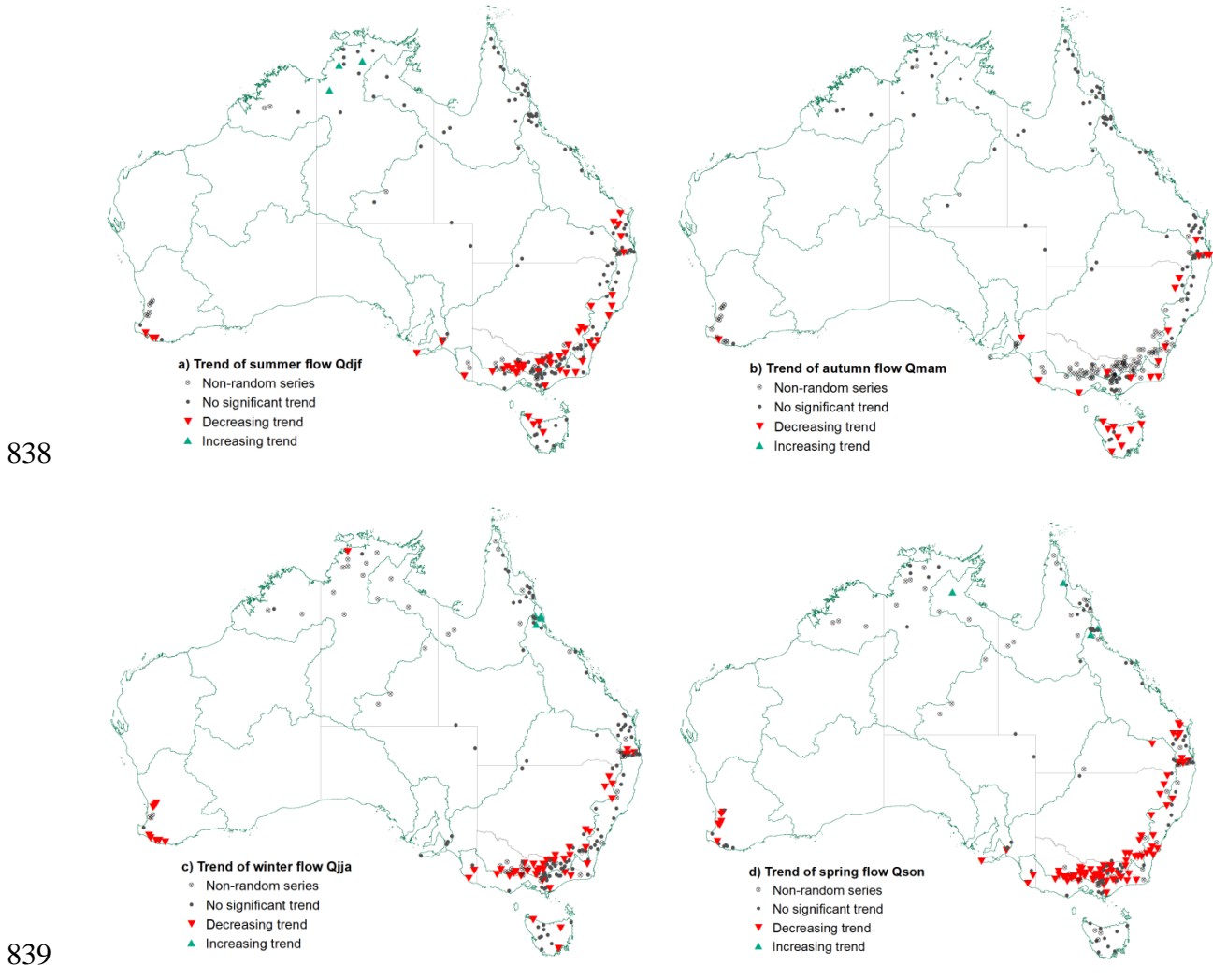



Figure 9: Maps showing trends of seasonal flow in a) $Q_{DJF}$ Summer flow; b) $Q_{MAM}$ Autumn flow; c) $Q_{JJA}$ Winter flow; d) $Q_{SON}$ Spring flow