# Peer review of "How streamflow has changed across Australia since the 1950's: evidence from the network of Hydrologic Reference"

_Hydrology and Earth System Sciences, 2015_

## Referee Comment (RC1) · Anonymous Referee #1 · 8 Mar 2016

Peer review for "How streamflow has changed across Australia since 1950's: evidence from the network of Hydrologic Reference Stations by S. X. Zhang et al. MS No.: hess-2015-464 MS Type: Research article

General comments

This article is well written overall and provides important results relevant to historical changes in Australian streamflows. It appears to be the first comprehensive analysis of streamflow trends and variability for Australia. It utilizes a newly available data set of minimally disturbed streamflow basins, which is critical for looking at climate driven changes. I don't see any major technical problems, however, more information is needed as it appears that catchments with different time periods are analyzed

together, this limits the comparability of catchments which is important for this type of trend/variability analysis. The results and discussion in places could be clarified, and better match the article figures. It's important to add at least a first cut at relating streamflow variability to large scale atmosphere/ocean patterns, particularly with the large number of step changes that were found.

Specific comments

Line 109, Was there a criteria for the Hydrologic Reference Stations for extensive basin water use or groundwater pumping? This could be hard to quantify, but is important, especially for low flows and in dry areas.

Line 109, Was any consideration given for catchments with substantial overlap in area (nested basins). Basins with substantial overlap would not offer independent information for an analysis.

Line 133, It's stated that "the primary data used in this study" are from the HRS network. Does this mean that stations outside the HRS network were used? This is problematic, if this is the case for this analysis.

Line 145, Could use more specifics on how well the model did for filling in data gaps; "perform well" is quite vague.

Line 159, I don't recall any discussion of the data collection agency/agencies. Were they collected by the same agency? If not, do they meet the same standards for inclusion in the HRS? If not, how do you assure consistency across regions when analyzing trends or variability? Have collection methods remained constant over time? This should be addressed. If not consistent over time, monotonic trends or step changes could be biased.

Line 173, Why isn't Qmin (for 1-day, 7-day or similar) analyzed? These low flows are typically important for water managers and ecological flows.

Line 186, Does the Median Crossing and Rank Difference test consider the possibility

of long-term persistence? If not, an important type of autocorrelation is being ignored.

Line 192, it doesn't appear that consistent periods of record were used for the various trend/step change tests in the article. This limits the comparability of results between catchments. Please provide more information. Authors should consider doing tests for selected periods and only including sites with mostly complete data for those periods. Multiple periods could be used, such as a 30 year period up to the present and a 50 year period up to the present. I don't recall a mention of what the last water year in this analysis is. This is important.

Line 194, Why not use the non-parametric Sen slope instead of least squares regression. Regression is sensitive to non-normality and outliers. Skewed distributions and outliers were noted previously in the article.

Line 251, The first sentence that summarizes trends seems inconsistent with the second sentence. Please reword.

Line 261, I think of trends as being one type of non-stationarity.

Line 261, Not clear what this paragraph is getting at, suggest expanding or contracting it.

Line 267, Need quick summary of trend methods.

Line 271, Suggest rewording, this statement seems incorrect. All stations showing significant trends are in the south (depending on how you define south) and all increasing trends are in the north.

Line 274. Why not test the importance of the last decade on trends? This could be done by repeating analyses but removing the last decade. This would be easy or hard, depending on how automated the trend testing is.

Line 275, Need Murray-Darling labeled on the figures and also the major regions of Australia (boundaries already in place for the major regions) for readers not from Aus-

tralia.

Line 280, Did you do trends in baseflow or baseflow index? The former is described in the methods and the latter is labeled in Table 2. The interpretation of these is obviously different.

Line 302, Why aren't the numerous step change decreases from the 1970s in south-eastern Australia (Figure 6) mentioned?

Line 306, Rainfall changes, whether they are monotonic trends or step changes would force streamflow changes. Please clarify.

Line 307, Please state what percentage of sites in different regions had significant Mann-Kendall trends, step changes, or both, and comment on whether, for the latter, this implies that Mann-Kendall significant trends were due to step changes.

Line 329, Why mention only winter trends for southern Australia, all seasons seem to have significant downward trends, with autumn having fewer than the others. Please clarify.

Line 358, Specify what parts of Australia these are here for non-Australians (to avoid people having to look for this earlier in the article).

Line 361, Rainfall deficiency "observed all over the continent" is not consistent with streamflow increases in the north.

Line 362, The accuracy of the statement on drought conditions depends on what type of drought you're referring to (meteorological, hydrological, soil moisture, etc.). This statement isn't correct if it refers to rainfall deficiencies, as those drive streamflow (not the reverse).

Line 368, need reference after "decade."

Line 370. It would be very useful, in helping to interpret trends (especially with the large number of step changes) to look at the relation between streamflow statistics

and major atmosphere/ocean patterns. A thorough analysis I can understand being beyond the scope of the article, but a first cut I think is reasonable and important. For example, you could correlate the interannual variability of streamflow statistics to major atmosphere/ocean indices. I'm not familiar with which ones are important for Australia, but ones that are known or suspected to be important to rainfall or streamflows could be tested. These could be relatively easy and may provide valuable information for interpreting the step changes. The discussion could also focus on the timing of known changes (what year) for indices that are important to Australian hydrology and compare those to the years that catchments showed step changes.

Line 396, It seems like the text describing trends for different regions doesn't match Figure 5. Rather than "Northern Territory and north-west of Western Australia, shouldn't it be "northern part of Northern Territory"? There's only one weak trend in northern Western Australia.

Line 401, Catchments in the southeast of S. Australia have significant downward trends in Figure 5.

Line 413, Both areas have a mix of step changes in the 1990s and 1970s in Figure 6.

General comment on figures: the trend symbols are too small in Figures 5-8.

Technical corrections and typos

Line 396, incorrect figure reference.

Figure 5 caption, change "decrease" to "decreasing"

[Figure]

---

## Short Comment (SC1) · 16 Mar 2016

As an Australian, I have read the submission with great interest and was pleased to see the analysis undertaken. I highlight the substantial amount of work that has gone into curating and making sense of such a large dataset at this scale. This is important progress and important not only for scientific purposes but for shaping policy in Australia. I would like to make a few short suggestions that could be considered during the discussion/revision process.

 c I think there are some problems with the section headings. Aside from the fact the sub-sub-heading is larger font than the sub-heading, I also note that section 4 is "Results and Discussion" and section 5 is "Discussion" ... There is also a Section 6 with

"Conclusions". I suggest these 3 sections and their sub-headings could be carefully looked at, and would suggest to split results and discussion into separate sections, with sub-headings used in the discussion to help navigate the reader to the significant findings.

• Further from the above, the aim as stated at the end of the introduction is to provide a nationwide assessment of trends in streamflow which is achieved well. Of course one of the powers of compiling the dataset is to the try tease out the science of why trends are occurring and it would be nice to see this as an aim. I notice a brief paragraph on this point (Page 13) highlighting general drying trend in the climate etc, but I felt the study would become much more powerful if there was a more significant attempt to explain the non-stationary behaviour. This could range from a quantitative assessment of changes in the rainfall-runoff coefficient (is the streamflow change amplifying or dampening the broad rainfall trends in each region?) or at a minimum could consist of a more detailed and focused discussion on Page 13 introducing and citing previous studies explaining mechanisms for the trends. For example, Smettem et al 2013 undertook an analysis on the forest response to drying trend impacts streamflow; Ummenhofer et al., 2009 on mechanisms for increasing drought; there are obviously many more papers relevant to different regions that could help readers understand the mechanism and significance of the trend. It is stated as being beyond the scope (in ln376), however, I would suspect it would be of key interest to the HESS readership and I would suggest that space could be made by moving section 3 and Figure 3 to an Appendix; in fact I would encourage the authors to refocus the aims on the hydrological trends AND their explanations, rather than the focus on the web portal itself.

• Lastly, whilst it is related to the above, it would be ideal for the discussion to cover the projections of climate change for the different regions to address the question of whether the the past changes are likely to continue, and as justification for the ongoing monitoring and assessment at the nation-wide scale. This need not be an extensive addition, just some targeted references cited for interested readers, potentially within a

dedicated sub-section in the discussion.

Thank you very much for the opportunity to comment on this great study, and I do hope these comments will be seen as constructive criticisms to help improve the overall paper and usefulness of the analysis.

---

## Referee Comment (RC2) · Anonymous Referee #2 · 29 Apr 2016

General comments The overall impression of this paper is that it is very clear, well-structured and interesting. The topic of temporal hydrologic change is highly relevant, and the quantitative data analysis of 222 stream gauges is comprehensive and previously unprecedented.

The paper presents a neat compilation of a large quantity of data and addresses relevant scientific questions within the scope of HESS – both regarding the issues of temporal hydrolgic change, but also the central question regarding aggregation, compilation and presentation of large data quantities (daily discharge values for 222 stations for ∼45 years).

The presentation of the HRS web portal great! This is a valuable resource, which will

be of great use for the international hydrological community. A paper such as "How streamflow has changed across Australia..." will (apart from its research significance in other ways) have an additional value of helping more researches find the publicly available Australian discharge data.

The paper is written in a clear, consise and straightforward manner, answering most questions that arise. The title clearly reflects the contents of the paper. The language is (as far as I can judge) fluent and correct, the paper is generally very readable. The mathematical formulae, symbols, abbreviations, and units correctly are correctly defined and used. The length of the paper is exemplary short, but still comprehensive enough.

The abstract provides a concise and complete summary, although I'm slightly confused about the expression 'living gauges'.

The scientific methods and assumptions are valid and clearly outlined, allowing reproduction (and traceability of results, as all data and used equations are publicly accessible). The statistical methods are thoroughly explained, and the decision to have these equations in an appendix is wise. The amount and quality of supplementary material is considered appropriate, and the figures and tables are generally in good shape, and are referred to accordingly.

In general, the number and quality of references seems appropriate for the topic, even though I think that a few more references regarding climate change could have been provided. Especially, I miss a reference to the most recent IPCC which would be of value here.

The scientific approach and the applied methods are valid and the results are to be sufficient to support the interpretations, and the substantial conclusions that are reached.

Specific comments My primary concern regards the limited reasoning regarding how the temporal change in streamflow is interrelated to a temporal change in precipitation.

The authors mention clearly that this is not within the scope of the study – which of course is fine. However, the dry period in the last decade in the south-eastern and south-western region is mentioned as a cause of some of the general downward trend. Although a thorough analysis is of course not viable within the scope of this paper, it would be nice to (if possible) have some discussion regarding the likeliness of this downward trend only being a consequence of the rainfall during a few dry years, or if the trend is likely to be consistent in the longer time perspective. Looking at table 2, at the years of the step change – 1996 is clearly the most dominating year (13 of 22!): an added reflection regarding the impacts of this (probably very non-normal) hydrological year would be interesting. How much impact does this "outlier year" have on the temporal trend? Would the same general pattern be seen even if it was to be omitted from the analysis? I do not request you to do the complete analysis of this issue, but some kind of (further) discussion on the topic could be useful.

Also, I believe that most data is available from the 1950's and onwards. However, I guess that longer time series should be available at least for some gauges. A comparison regarding an even more long-term time series would give additional weight to the results – although, this may be the subject of another study.

Line 152 – please also add the median time-series length.

Lines 206-208 – is any of this presented here? Or mainly as background info to the tables/figures?

Line 262 – shouln't also land-use changes be mentioned in this context?

One last comment: the fact that different hydrologic years are used for different stations (if I understand it correct) – will this have an impact on the results (lines 149-151)?

Technical corrections There are hardly any technical corrections that need to be addressed in the paper. The authors have made a robust study, and compiled the data in a presentable and concise manner.
I am however not clear about what the authors mean by the concept of 'living gauges', neither in the abstract nor in the text (lines 29 and 93) – don't just normal gauges record and detect changes in hydrologic responses?

As not being very familiar with Australian geography, I would have appreciated (if possible to do in an aesthetic manner) information regarding the names of the basins in figure 1 – perhaps by inserting the roman numerals from table 1 on the map?

Also, table 2 seems to be of somewhat low resolution (the letters are blurry) – if possible, please improve this.

Figure 5 (and 6 and 8), please add Q_(appropriate index) in the text for clarity.

Thanks for a good read, and congratulations on your thorough study! I'm looking forward to seeing more of this paper in the future!

---

## Author Comment (AC1) · 27 May 2016

(Short comments from M. Hipsey, matt.hipsey@uwa.edu.au,)

**Authors' response to Short comments from M. Hipsey**

17 May 2016
S. X. Zhang et al.
Sophie.Zhang@bom.gov.au

Response to review "Short Comments": Thank you for your time to review our manuscript. You have mentioned valuable points, which we really appreciate. Please find our response below to your comments, questions and suggestions. The referee's comments are first recalled in *italics, blue colour font*, and then followed by our answer.

*As an Australian, I have read the submission with great interest and was pleased to see the analysis undertaken. I highlight the substantial amount of work that has gone into curating and making sense of such a large dataset at this scale. This is important progress and important not only for scientific purposes but for shaping policy in Australia.*
*I would like to make a few short suggestions that could be considered during the discussion/revision process.*
*I think there are some problems with the section headings. Aside from the fact the sub-sub-heading is larger font than the sub-heading, I also note that section 4 is*
*"Results and Discussion" and section 5 is "Discussion" ... There is also a Section 6 with "Conclusions". I suggest these 3 sections and their sub-headings could be carefully*
*looked at, and would suggest to split results and discussion into separate sections, with sub-headings used in the discussion to help navigate the reader to the significant findings.*
**Answer**
Thank you for noticing this and helpful suggestions to improve the structure of manuscript. We will adjust all the headings at different levels in a systematic way to reflecting the hierarchy structure clearly. Also the section or sub-section titles will be modified, in the way to keep "Results and Discussion" while merge "Discussion" into it, and as suggested, with sub-headings used in the discussion to help navigate the reader to the significant findings.

*Further from the above, the aim as stated at the end of the introduction is to*
*provide a nationwide assessment of trends in streamflow which is achieved well. Of*
*course one of the powers of compiling the dataset is to the try tease out the science*
*of why trends are occurring and it would be nice to see this as an aim. I notice a brief*
*paragraph on this point (Page 13) highlighting general drying trend in the climate etc,*
*but I felt the study would become much more powerful if there was a more significant*
*attempt to explain the non-stationary behaviour. This could range from a quantitative*

assessment of changes in the rainfall-runoff coefficient (is the streamflow change amplifying or dampening the broad rainfall trends in each region?) or at a minimum could consist of a more detailed and focused discussion on Page 13 introducing and citing previous studies explaining mechanisms for the trends. For example, Smettem et al 2013 undertook an analysis on the forest response to drying trend impacts streamflow; Ummenhofer et al., 2009 on mechanisms for increasing drought; there are obviously many more papers relevant to different regions that could help readers understand the mechanism and significance of the trend. It is stated as being beyond the scope (in ln376), however, I would suspect it would be of key interest to the HESS readership and I would suggest that space could be made by moving section 3 and Figure 3 to an Appendix; in fact I would encourage the authors to refocus the aims on the hydrological trends AND their explanations, rather than the focus on the web portal itself.

**Answer**

We agree to this point. Though a thorough investigation of reasons behind the hydrological trends is beyond scope of this article, we added relevant literatures on past climate changes, non-stationarity in streamflow Australia (including the papers you mentioned - Smettem et al 2013; Ummenhofer et al., 2009), and extend the discussion accordingly, also to relate the flow changes with rainfall.

An example for that, adding a trend map of rainfall for discussion. The Figure below gives an example showing an updated summary of long-term rainfall trends (1950-2015). Changes in precipitation or other climate variables impact on the rainfall-runoff process directly, and indirectly causing changes in flora, relief and soil erosion. The identified trend patterns in annual total streamflow are spatially consistent with trends in annual total rainfall as shown in this Figure, where most of eastern and south-western Australia has experienced substantial rainfall declines since 1950; while north-western Australia has become wetter over this period. This similarity implies that hydrological variability is closely related with changes in rainfall patterns.

[Figure]

(source: http://www.bom.gov.au/climate/change/index.shtml#tabs=Tracker&tracker=trend-maps&tQ%5Bmap%5D=rain&tQ%5Barea%5D=aus&tQ%5Bseason%5D=0112&tQ%5Bperiod%5D=1950)

About section 3 on web portal development, it provides important information for key users of this study. It will be kept in the main text however we will make it concise.

**Answer**
This study is focused on the flow changes that we observed from the historical data records, and not trying to refer it to the future. From the sustainable yield study done by CSIRO (http://www.csiro.au/en/Research/LWF/Areas/Water-resources/Assessing-water-resources/Sustainable-yields ), It is likely that rainfall/streamflow trends in  the Murray-Darling Basin, southern Australia and south-west Western Australia is likely to continue.
The suggestion you raised here will be an interesting point to look at, but unfortunately it's out of scope of this paper. We added relevant literatures on past climate changes (as mentioned above), and included discussion on relating the flow changes with rainfall trend, which we hope will provide some useful information for readers to understand and interpret the trends and step changes presented.

Thanks again for your valuable comments!

---

## Author Comment (AC2) · 27 May 2016

(Anonymous Referee #1,)

**Authors' response to Referee #1**

17 May 2016
S. X. Zhang et al.
Sophie.Zhang@bom.gov.au

Response to review referee #1: Many thanks for your time to review our manuscript. We highly appreciate your insightful and constructive comments which will help to improve the submitted manuscript. Please find our response below to your comments, questions and suggestions. The referee's comments are first recalled in *italics, blue colour font*, and then followed by our answer.

*General comments*

*This article is well written overall and provides important results relevant to historical changes in Australian streamflows. It appears to be the first comprehensive analysis of streamflow trends and variability for Australia. It utilizes a newly available data set of minimally disturbed streamflow basins, which is critical for looking at climate driven changes. I don't see any major technical problems, however, more information is needed as it appears that catchments with different time periods are analysed together, this limits the comparability of catchments which is important for this type of trend/variability analysis. The results and discussion in places could be clarified, and better match the article figures. It's important to add at least a first cut at relating streamflow variability to large scale atmosphere/ocean patterns, particularly with the large number of step changes that were found.*

**Answer**
The authors would like to thank the referee for those positive evaluations of the manuscript and our work; and for the insightful comments on the data and method.

For the first question in general comments: catchments with different time periods are analysed together. Firstly, we checked the statistics of data availability of all 222 stations, with minimum 30 years data, average 48 years data, and 86% stations longer than 40 years data. Most of the stations have observations starting in 1970s. One intention of this study is to look at the long term changes in Australian streamflow, therefore the full length of observations of all stations are used in the analysis. If the data is truncated to have a consistent time periods over the continent, this means only 30 years data to be used, and for some stations half of data length will be cut which will limits an holistic examination of the historic records. Secondly, the data length of every station was not exactly the same over the continent, but for the stations within the same region, the data lengths were in more consistent time periods. And data in most of stations (86%) has very similar time period. These make the comparisons on a fairly consistent base.

Lastly, the primary purpose of the work is to provide long-term trends of streamflow data of Australia rivers with observation data as much as available. This helps to  provide long-term high-quality daily flow observations data of Australian rivers for a broad range of users and researchers. We might be able to try the continental scale analysis within a common period as the next project activity.

For the second question in general comments: relating streamflow variability to large scale atmosphere/ocean patterns. We agree to this point. Though a thorough analysis of the relationship between streamflow and climate indices is out of scope of this study, we have added relevant literatures on climate, and include a discussion to relate the flow changes with main climate indices. To show an example for that: adding a trend map of rainfall for discussion. The Figure below gives an example showing an updated summary of long-term rainfall trends (1950-2015). Changes in precipitation or other climate variables impact on the rainfall-runoff process directly, and indirectly causing changes in flora, relief and soil erosion. The identified trend patterns in annual total streamflow are spatially consistent with trends in annual total rainfall as shown in this Figure, where most of eastern and south-western Australia has experienced substantial rainfall declines since 1950; while north-western Australia has become wetter over this period. This similarity implies that hydrological variability is closely related with changes in rainfall patterns.

[Figure]

(source: http://www.bom.gov.au/climate/change/index.shtml#tabs=Tracker&tracker=trend-maps&tQ%5Bmap%5D=rain&tQ%5Barea%5D=aus&tQ%5Bseason%5D=0112&tQ%5Bperiod%5D=1950)

*Specific comments*

*Line 109, Was there a criteria for the Hydrologic Reference Stations for extensive basin water use or groundwater pumping? This could be hard to quantify, but is important, especially for low flows and in dry areas.*

**Answer**

Yes, whether there is water use diversion or not, is important especially for low flow or dry areas. There was a criterion regarding this: line 109 "unregulated catchments with minimal land use change". Catchments with extensive basin water use or groundwater pumping were filtered and not included in HRS catchments, based on the local knowledge of the basin, stakeholder consultation and land use change analysis. For more information on the station selection process and involvement of stakeholders in prioritising the stations list, users can refer to the HRS web site at http://www.bom.gov.au/water/hrs/guidelines.shtml .

*Line 109, Was any consideration given for catchments with substantial overlap in area (nested basins). Basins with substantial overlap would not offer independent information for an analysis.*

**Answer**

Overlapping in area has been considered. We have checked all HRS stations thoroughly, only at a few locations there are nested basins: 3 in Queensland, 2 in North Territory and 1 in New South Wales. 5 of them have less than 10% area overlapping (which can be considered as independent) and only 1 has 50%. The influence to overall analysis is marginal.

*Line 133, It's stated that "the primary data used in this study" are from the HRS network.*
*Does this mean that stations outside the HRS network were used? This is problematic, if this is the case for this analysis.*

**Answer**

All data used are from the HRS network. We have edited this sentence by removing "primary", and have "all data used" instead.

*Line 145, Could use more specifics on how well the model did for filling in data gaps; "perform well" is quite vague.*

**Answer**

We agree and have specifide the model performance in this part, with the statistics of NSE results: Median = 0.74; mean 0.72; STDEV =0.12

*Line 159, I don't recall any discussion of the data collection agency/agencies. Were they collected by the same agency? If not, do they meet the same standards for inclusion in the HRS? If not, how do you assure consistency across regions when analysing trends or variability? Have collection methods remained constant over time? This should be addressed. If not consistent over time, monotonic trends or step changes could be biased.*

**Answer**

Water data is collected across Australia by many organisations, utilities and regulators in different states and territories, often to meet the requirements of their own documented procedures and sometimes with reference to Australian or international standards or guidelines. The Bureau's role as the national water information provider, has been working collaboratively with the water industry to develop and promote water information standards and guidelines to collate, interpret and access nationally consistent data. All data included in the HRS database are compiled, quality-checked by the Bureau, and therefore are consistent nationally and over the time. Bureau has developed a set of standard data quality code and

references guides on how it relates to different agencies quality code. This will be addressed in the manuscript.

*Line 173, Why isn't Qmin (for 1-day, 7-day or similar) analyzed? These low flows are typically important for water managers and ecological flows.*

**Answer**

Thanks for suggesting that. Qmin was not included in the current work, but it's a good idea and it can be considered in the future.

*Line 186, Does the Median Crossing and Rank Difference test consider the possibility of long-term persistence? If not, an important type of autocorrelation is being ignored.*

**Answer**

Yes these 2 tests (Median crossing & Rank difference) consider the long-term persistence as well (Kundzewicz and Robson, 2000). Autocorrelation checking was part of the randomness test, so it was not missed in this analysis.

*Line 192, it doesn't appear that consistent periods of record were used for the various trend/step change tests in the article. This limits the comparability of results between catchments. Please provide more information. Authors should consider doing tests for selected periods and only including sites with mostly complete data for those periods.*
*Multiple periods could be used, such as a 30 year period up to the present and a 50 year period up to the present. I don't recall a mention of what the last water year in this analysis is. This is important.*

**Answer**

The first part of this comment is addressed above in general comments. For the second question, the data used in this study are up to end of 2014, so the last water year is 2014. We have added this in the text accordingly.

*Line 194, Why not use the non-parametric Sen slope instead of least squares regression.*
*Regression is sensitive to non-normality and outliers. Skewed distributions and outliers were noted previously in the article.*

**Answer**

Many thanks for this suggestion. We agree completely as the non-parametric Sen slope is more appropriate method for this data set than the least square regression method which is parametric. We have followed the advice to apply the non-parametric Sen slope instead of LSR, and update the results in Table 2. The following are the references for non-parametric slope estimator: Sen, Pranab Kumar (1968, Journal of the American Statistical Association 63: 1379–1389), Theil, H. (1950), which were added to the paper. The Sen slope method was not in the current system, however, it will be considered in the next HRS web portal upgrade.

*Line 251, The first sentence that summarizes trends seems inconsistent with the second sentence. Please reword.*

**Answer**

The two sentences of this paragraph are rephrased to make the idea clearly, in this way: "Patterns of trends were noted in the different flow regimes. Moving through the flow variables from low (Q10), to median (Q50), to high (Q90), and onto maximum (Qmax), an increasing number of stations were found with no trends, combined with decreasing number for non-random series."

*Line 261, I think of trends as being one type of non-stationarity.*

**Answer**

We agree that trend or step change is one type of non-stationarity. Text is reworded in such a way that trend and step-change come under ' non-stationarity'.

*Line 261, Not clear what this paragraph is getting at, suggest expanding or contracting it.*
**Answer**
We tried to address the question broadly, what could be the reasons behind the observed flow changes, as an introducing paragraph for the following sections of trend and step changes. Possible reasons are: (i) changes in rainfall resulted in changes in streamflow – rainfall-runoff process has changed; (ii) distribution of rainfall within the year, (iii) changes in unconfined groundwater level, particularly in the streamzone, changes in threshold, (iv) probable changes in evapotranspiration process due to though no significant changes in landuse/landcover.

This paragraph has been rephrased accordingly, such as "Detecting the trend and non-stationarity in a hydrologic time series may help us to understand the possible links between hydrological processes and global environment changes. Many of the streamflow time series in this dataset exhibit trends or step-changes in the mean or median. Abrupt changes and trends in the hydrologic time series could be indicators of hydrologic non-stationarity or long-term gradual changes in the rainfall-runoff transformation processes."

*Line 267, Need quick summary of trend methods.*
**Answer**
A quick summary of trend methods was added here, to support the results statement.

*Line 271, Suggest rewording, this statement seems incorrect. All stations showing significant trends are in the south (depending on how you define south) and all increasing trends are in the north.*
**Answer**
We have reworded the sentence, such as "all stations showing decreasing trends (35% of stations) are in the southern part of Australia and all stations showing increasing trends (4% of stations) in the northern part".

*Line 274. Why not test the importance of the last decade on trends? This could be done by repeating analyses but removing the last decade. This would be easy or hard, depending on how automated the trend testing is.*
**Answer**
This will be interesting to look at, but it is not within the scope of the present paper. We would prefer to keep the current trend testing results, and put the suggestion as future research work.

*Line 275, Need Murray-Darling labeled on the figures and also the major regions of Australia (boundaries already in place for the major regions) for readers not from Australia.*
**Answer**
Figure 1, 5, 6 have been modified, with the basin code (from I to XII ) marked for each region, and readers can refer the basin names to Table1.

*Line 280, Did you do trends in baseflow or baseflow index? The former is described in the methods and the latter is labeled in Table 2. The interpretation of these is obviously different.*
**Answer**
Thank you for pointing out this. The trend test was applied to baseflow, not baseflow index. In Table 2, baseflow index was listed there (calculated by the ratio of baseflow to total flow), and the trend results of baseflow was indicated at the top right corner.

*Line 302, Why aren't the numerous step change decreases from the 1970s in southeastern Australia (Figure 6) mentioned?*

**Answer**

Discussion is added, in section 4, sub-section about "step change", to address the step change decreases from the 1970s in south-eastern Australia.

*Line 306, Rainfall changes, whether they are monotonic trends or step changes would force streamflow changes. Please clarify.*

**Answer**

More discussions were added here for relating rainfall changes with flow trends, as addressed above in general comments.

*Line 307, Please state what percentage of sites in different regions had significant Mann-Kendall trends, step changes, or both, and comment on whether, for the latter, this implies that Mann-Kendall significant trends were due to step changes.*

**Answer**

We have added a quick summary of result statistics for trends and step changes for different regions.

*Line 329, Why mention only winter trends for southern Australia, all seasons seem to have significant downward trends, with autumn having fewer than the others. Please clarify.*

**Answer**

More discussion was added here to cover more aspects of seasonal changes.

*Line 358, Specify what parts of Australia these are here for non-Australians (to avoid people having to look for this earlier in the article).*

**Answer**

Texts were revised to specify what parts of Australia. Also Figure 1, 5, 6 were modified with the basin code, to make it more clear where the discussion is about (it's addressed above in Line 275 question).

*Line 361, Rainfall deficiency "observed all over the continent" is not consistent with streamflow increases in the north.*

**Answer**

This sentence was rephrased accordingly.

*Line 362, The accuracy of the statement on drought conditions depends on what type of drought you're referring to (meteorological, hydrological, soil moisture, etc.). This statement isn't correct if it refers to rainfall deficiencies, as those drive streamflow (not the reverse).* **Answer**

Accuracy of statement on drought was added, with referring to literature on the severe drought in southeast Australia 1997-2009 (SEACI, 2011, The Millennium Drought and 2010/11 Floods; Ummenhofer et.al, 2009).

*Line 368, need reference after "decade."*

**Answer**

Reference was added: decade means for the years 1997–2009 inclusive. SEACI, 2011, The Millennium Drought and 2010/11 Floods, http://www.seaci.org/publications/documents/SEACI-2Reports/SEACI2_Factsheet2of4_WEB_110714.pdf.

*Line 370. It would be very useful, in helping to interpret trends (especially with the large number of step changes) to look at the relation between streamflow statistics and major atmosphere/ocean patterns. A thorough analysis I can understand being beyond the scope of the article, but a first cut I think is reasonable and important. For example, you could correlate the interannual variability of streamflow statistics to major atmosphere/ocean indices. I'm not familiar with which ones are important for Australia, but ones that are known or suspected to be important to rainfall or streamflows could be tested. These could be relatively easy and may provide valuable information for interpreting the step changes. The discussion could also focus on the timing of known changes (what year) for indices that are important to Australian hydrology and compare those to the years that catchments showed step changes.*

**Answer**

This is already addressed above in general comments, more discussion were added for this point.

*Line 396, It seems like the text describing trends for different regions doesn't match Figure 5. Rather than "Northern Territory and north-west of Western Australia, shouldn't it be "northern part of Northern Territory"? There's only one weak trend in northern Western Australia.*

**Answer**

We have modified the text to be more specifically describing the locations, such as "In Northern Territory there was an increasing trend in annual streamflow (QT) while there was no significant trend visible in the northern region of Queensland. However, in south-eastern Queensland there was a significant decreasing trend. Most of the gauging stations in New South Wales, Victoria, south-east South Australia, south-west Western Australia, and north-west Tasmania showed a significant decreasing trend in annual streamflow. In South Tasmania, central Australia, most of the stations showed no significant trend in annual streamflow. "

*Line 401, Catchments in the southeast of S. Australia have significant downward trends in Figure 5.*

**Answer**

We have included it in the text.

*Line 413, Both areas have a mix of step changes in the 1990s and 1970s in Figure 6.*
*General comment on figures: the trend symbols are too small in Figures 5-8.*
*Technical corrections and typos*

**Answer**

We have added this comment in the paragraph, and improve the figure quality (enlarge the symbol, size of graph, and improve the resolution).

*Line 396, incorrect figure reference.*

**Answer**

It's been corrected as "Figure 1". Thanks for pointing it out.

*Figure 5 caption, change "decrease" to "decreasing"*

**Answer**

We have changed the caption accordingly.

Thank you again for your valuable comments!

---

## Author Comment (AC4) · 27 May 2016

Response to review referee #2: Many thanks for the positive evaluation of our work. We highly appreciate your explicit comments and suggestions which will help to improve the submitted manuscript. Please find our response below to your comments, questions and suggestions. The referee's comments are first recalled in italics, blue colour font, and then followed by our answer.

Please also note the supplement to this comment:
http://www.hydrol-earth-syst-sci-discuss.net/hess-2015-464/hess-2015-464-AC4-supplement.pdf

---

## Author Response (AR1)

**Reply to the comments for revised manuscript "How streamflow has changed across Australia since the 1950's: evidence from the network of Hydrologic Reference Stations" by S. X. Zhang et al.**

**Authors' point-by-point response to all comments received**

August 2016
S. X. Zhang et al.
Sophie.Zhang@bom.gov.au

**1. Authors' response and changes to review Referee #1,**

(Received and published 8 March 2016; Editor's decision to compile all the changes and upload the revised manuscript, received 30 June 2016)

Many thanks for your time to review our manuscript. We highly appreciate your insightful and constructive comments which will help to improve the submitted manuscript. Please find our response below to your comments, questions and suggestions. The referee's comments are first recalled in *italics, blue colour font*, and then followed by our **response** and **changes** made in the revised manuscript.

**General comments**

*This article is well written overall and provides important results relevant to historical changes in Australian streamflows. It appears to be the first comprehensive analysis of streamflow trends and variability for Australia. It utilizes a newly available data set of minimally disturbed streamflow basins, which is critical for looking at climate driven changes. I don't see any major technical problems, however, more information is needed as it appears that catchments with different time periods are analysed together, this limits the comparability of catchments which is important for this type of trend/variability analysis. The results and discussion in places could be clarified, and better match the article figures. It's important to add at least a first cut at relating streamflow variability to large scale atmosphere/ocean patterns, particularly with the large number of step changes that were found.*

**Answer**

The authors would like to thank the referee for those positive evaluations of the manuscript and our work; and for the insightful comments on the data and method.

For the first question in general comments: catchments with different time periods are analysed together. Firstly, we checked the statistics of data availability of all 222 stations, with minimum 30 years data, average 48 years data, and 86% stations longer than 40 years data. Most of the stations have observations starting in 1970s. One intention of this study is to look at the long term changes in Australian streamflow, therefore the full length of observations of all stations are used in the analysis. If the data is truncated to have a consistent time periods over the continent, this means only 30 years data to be used, and for some stations half of data length will be cut which will limits an holistic examination of the historic records.

Secondly, the data length of every station was not exactly the same over the continent, but for the stations within the same region, the data lengths and observation time periods were more consistent. Additionally, data of stations (86%) has very similar time period which implies that the comparisons are mostly consistent.

Lastly, the primary purpose of the work is to provide long-term trends of streamflow data of
Australian rivers with as much observation data as available. This helps to provide long-term
high-quality daily flow observations data of Australian rivers for a broad range of users and
researchers. In our next biennial upgrade of the Hydrologic Reference web portal, we will try
and include the continental scale analysis within a common period as an important project
activity.
For the second question in general comments: relating streamflow variability to large scale
atmosphere/ocean patterns. We agree to this point. Though a thorough analysis of the
relationship between streamflow and climate indices is out of scope of this study, we have
added relevant literatures on climate, and include a discussion (see section 5.2 in the revised
manuscript) to relate the flow changes with main climate indices. We illustrate the rainfall
trend map for discussion. The figure below (Figure 1) gives an example showing an updated
summary of long-term rainfall trends (1970-2015). Changes in precipitation or other climate
variables impact on the rainfall-runoff process directly, and indirectly causing changes in
flora, relief and soil erosion. The identified trend patterns in annual total streamflow are
spatially consistent with trends in annual total rainfall as shown in this Figure, where most of
eastern and south-western Australia has experienced substantial rainfall declines since
1970; while north-western Australia has become wetter over this period. This similarity
implies that hydrological variability is closely related with changes in rainfall patterns.
The intent of this paper is to quantify for the first time, trends in long-term streamflow data
across pristine catchments using a consistent methodology. We examine the non-stationarity
of the hydrologic response through a separate paper, currently under review (Ajami et al.;
under review with HESS).
The figure below was not attached in the revised manuscript but the website link was
provided.
**Changes:**
•   Added references on climate change, non-stationarity, and text revised accordingly
•   Added discussion in Section 5.2

[Figure]

Figure 1: Trend in annual total rainfall in Australia 1970-2015.
(Source: http://www.bom.gov.au/climate/change/#tabs=Tracker&tracker=trend-maps)

*Specific comments*
*Line 109, Was there a criteria for the Hydrologic Reference Stations for extensive basin water use or groundwater pumping? This could be hard to quantify, but is important, especially for low flows and in dry areas.*

**Answer**
Yes, whether there is water use diversion or not, is important especially for low flow or dry areas. There was a criterion regarding this: line 109 "unregulated catchments with minimal land use change". Catchments with extensive basin water use or groundwater pumping were filtered and not included in HRS catchments, based on the local knowledge of the basin, stakeholder consultation and land use change analysis. For more information on the station selection process and involvement of stakeholders in prioritising the stations list, users can refer to the HRS web site at http://www.bom.gov.au/water/hrs/guidelines.shtml .

**Changes:**
- Added explanation in section 2.1, line 113-115 (track-change version)

*Line 109, Was any consideration given for catchments with substantial overlap in area*
*(nested basins). Basins with substantial overlap would not offer independent information for*
*an analysis.*
**Answer**
Overlapping in area has been considered. We have checked all HRS stations thoroughly,
only at a few locations there are nested sub-basins: 3 in Queensland, 2 in North Territory and
1 in New South Wales. 5 of them have less than 10% area overlapping (which can be
considered as independent) and only 1 has 50%. The influence on overall analysis is
negligible.
*Line 133, It's stated that "the primary data used in this study" are from the HRS network.*
*Does this mean that stations outside the HRS network were used? This is problematic, if this*
*is the case for this analysis.*
**Answer**
All data used are from the HRS network. We have edited this sentence by removing
"primary", and have "all data used" instead.
**Changes:**
• Text changed in section 2.1, line 137 (track-change version)
*Line 145, Could use more specifics on how well the model did for filling in data gaps;*
*"perform well" is quite vague.*
**Answer**
We agree and have specified the model performance in this part, with the statistics of NSE
results: Median = 0.74; mean 0.72; STDEV =0.12
**Changes:**
• model performance specified in section 2.1, line 150-152 (track-change version)
*Line 159, I don't recall any discussion of the data collection agency/agencies. Were they*
*collected by the same agency? If not, do they meet the same standards for inclusion in the*
*HRS? If not, how do you assure consistency across regions when analysing trends or*
*variability? Have collection methods remained constant over time? This should be*
*addressed. If not consistent over time, monotonic trends or step changes could be biased.*
**Answer**
Water data is collected across Australia by many organisations, utilities and regulators in
different states and territories, often to meet the requirements of their own documented
procedures and sometimes with reference to Australian or international standards or
guidelines. The Bureau's role as the national water information provider, has been working
collaboratively with the water industry to develop and promote water information standards
and guidelines to collate, interpret and access nationally consistent data. All data included in
the HRS database are compiled, quality-checked by the Bureau, and therefore are consistent
nationally and over the time. Bureau has developed a set of standard data quality code and
references guides on how it relates to different agencies quality code. This has now been
addressed in the manuscript.
**Changes:**
• text added in section 2.1, line 173-181 (track-change version)
*Line 173, Why isn't Qmin (for 1-day, 7-day or similar) analyzed? These low flows are typically*
*important for water managers and ecological flows.*
**Answer**
Thanks for suggesting that. Qmin was not included in the current work, but it's a good idea
and will include it in our future work.

*Line 186, Does the Median Crossing and Rank Difference test consider the possibility of*
*long-term persistence? If not, an important type of autocorrelation is being ignored.*
**Answer**
Yes these 2 tests (Median crossing & Rank difference) consider the long-term persistence as
well (Kundzewicz and Robson, 2000). Autocorrelation checking was part of the randomness
test, and therefore it was not missed in this analysis.
**Changes:**
• text added in section 2.3, line 212 (track-change version)
*Line 192, it doesn't appear that consistent periods of record were used for the various*
*trend/step change tests in the article. This limits the comparability of results between*
*catchments. Please provide more information. Authors should consider doing tests for*
*selected periods and only including sites with mostly complete data for those periods.*
*Multiple periods could be used, such as a 30 year period up to the present and a 50 year*
*period up to the present. I don't recall a mention of what the last water year in this analysis is.*
*This is important.*
**Answer**
The first part of this comment is addressed above in general comments. For the second
question, the data used in this study are up to end of 2014, so the last water year is 2014.
We have added this in the text accordingly.
**Changes:**
• text added in section 2.1, line 165 (track-change version)
*Line 194, Why not use the non-parametric Sen slope instead of least squares regression.*
*Regression is sensitive to non-normality and outliers. Skewed distributions and outliers were*
*noted previously in the article.*
**Answer**
Many thanks for this suggestion. We have followed the advice to apply the non-parametric
Sen Slope instead of LSR, and update the results in Table 2. The results show not much
different from the LSR method. The following are the references for non-parametric slope
estimator: Sen, Pranab Kumar (1968, Journal of the American Statistical Association 63:
1379–1389), Theil, H. (1950), which was added to the paper.
**Changes:**
• Updated Table 2 using Sen Slope
• text revised accordingly in Section 2.3
*Line 251, The first sentence that summarizes trends seems inconsistent with the second*
*sentence. Please reword.*
**Answer**
The two sentences of this paragraph are rephrased for clarity.
**Changes:**
• sentences rephrased in Section 4.1, line 280-283: "Patterns of trends were noted in
different flow regimes. Moving through the flow variables from low (Q10), to median
(Q50), to high (Q90), and onto maximum (Qmax), an increasing number of stations
were found with no trends, combined with decreasing number for non-random series.
"
*Line 261, I think of trends as being one type of non-stationarity.*
**Answer**
We agree that trend or step change is one type of non-stationarity. Text is reworded in such
a way that trend and step-change come under 'non-stationarity'.
**Changes:**

•   text revised accordingly in Section 4.2, line 296-300 (track-change version)

•   references on non-stationarity added

*Line 261, Not clear what this paragraph is getting at, suggest expanding or contracting it.*

**Answer**

We tried to address the question broadly, what could be the reasons behind the observed flow changes, as an introducing paragraph for the following sections of trend and step changes. This paragraph has been rephrased accordingly.

**Changes:**

•   text revised in Section 4.2, line 300-308 (track-change version)

*Line 267, Need quick summary of trend methods.*

**Answer**

A quick summary of trend methods was added here, to support the results statement.

**Changes:**

•   summary of trend methods was added in Section 4.2.1, line 311-317 (track-change version)

*Line 271, Suggest rewording, this statement seems incorrect. All stations showing significant*

*trends are in the south (depending on how you define south) and all increasing trends are in*

*the north.*

**Answer**

We have reworded the sentence.

**Changes:**

•   statement reworded in Section 4.2.1, line 320-323 (track-change version)

*Line 274. Why not test the importance of the last decade on trends? This could be done by*

*repeating analyses but removing the last decade. This would be easy or hard, depending on*

*how automated the trend testing is.*

**Answer**

This will be interesting to look at, but it is not within the scope of the present paper. We would prefer to keep the current trend testing results, and put the suggestion as future research work.

*Line 275, Need Murray-Darling labeled on the figures and also the major regions of Australia*

*(boundaries already in place for the major regions) for readers not from Australia.*

**Answer**

Figures have been modified.

**Changes:**

•   Figure 1 was modified with drainage division basin names (green colour font matching the colour of drainage division borders) labelled on the map

•   Figure 5 was modified with Australian state names (grey colour font matching the colour of Australian state borders) labelled on the map

*Line 280, Did you do trends in baseflow or baseflow index? The former is described in the*

*methods and the latter is labeled in Table 2. The interpretation of these is obviously different.*

**Answer**

Thank you for pointing out this. The trend test was applied to baseflow, not baseflow index.

In Table 2, baseflow index was listed there (calculated by the ratio of baseflow to total flow), and the trend results of baseflow was indicated at the top right corner.

**Changes:**

•     Clarification added in Section 4.2.1, line 332-333 (track-change version)

*Line 302, Why aren't the numerous step change decreases from the 1970s in southeastern*
*Australia (Figure 6) mentioned?*
**Answer**
Discussion is added, in section 4, sub-section about "step change", to address the step
change decreases from the 1970s in south-eastern Australia.
**Changes:**
•     Text added in Section 4.2.2, line 360-362 (track-change version)

*Line 306, Rainfall changes, whether they are monotonic trends or step changes would force*
*streamflow changes. Please clarify.*
**Answer**
More discussions were added here for relating rainfall changes with flow trends, as
addressed above in general comments.
**Changes:**
•     Discussion added in Section 5.2

*Line 307, Please state what percentage of sites in different regions had significant*
*Mann-Kendall trends, step changes, or both, and comment on whether, for the latter, this*
*implies that Mann-Kendall significant trends were due to step changes.*
**Answer**
We have added a quick summary of result statistics for trends and step changes for different
regions.
**Changes:**
•     result statistics for different regions added in Section 4.2.2, line 371-383 (track-
change version)
•     A new figure was added for this statistics summary, as Figure 7. Numbering of other
figures was changed accordingly.

*Line 329, Why mention only winter trends for southern Australia, all seasons seem to have*
*significant downward trends, with autumn having fewer than the others. Please clarify.*
**Answer**
Clarification was added here to cover more aspects of seasonal changes.
**Changes:**
•     Sentence reworded in Section 4.2.2, line 402-406 (track-change version)

*Line 358, Specify what parts of Australia these are here for non-Australians (to avoid people*
*having to look for this earlier in the article).*
**Answer**
Texts were revised to specify what parts of Australia. Also Figure 1 and 5 were modified with
the basin code, to make it more clear where the discussion is about (it's addressed above in
Line 275 question).
**Changes:**
•     Text revised in Section 5.1 (track-change version)
•     Figure 1 and 5 labelled with drainage basin name and Australian state name

*Line 361, Rainfall deficiency "observed all over the continent" is not consistent with*
*streamflow increases in the north.*
**Answer**
This sentence was rephrased accordingly.

**Changes:**

- Text revised in Section 5.2, line 485-486 (track-change version)

*Line 362, The accuracy of the statement on drought conditions depends on what type of drought you're referring to (meteorological, hydrological, soil moisture, etc.).*

**Answer**

Accuracy of statement on drought was added, with referring to literature on the severe drought in southeast Australia 1997-2009 (SEACI, 2011, The Millennium Drought and 2010/11 Floods; Ummenhofer et.al, 2009).

**Changes:**

- Text revised in Section 5.2, line 486-489 (track-change version)

*Line 368, need reference after "decade."*

**Answer**

Reference was added: decade means for the years 1997–2009 inclusive. SEACI, 2011, The Millennium Drought and 2010/11 Floods, http://www.seaci.org/publications/documents/SEACI-2Reports/SEACI2_Factsheet2of4_WEB_110714.pdf.

**Changes:**

- Literature added in References

*Line 370. It would be very useful, in helping to interpret trends (especially with the large number of step changes) to look at the relation between streamflow statistics and major atmosphere/ocean patterns. A thorough analysis I can understand being beyond the scope of the article, but a first cut I think is reasonable and important. For example, you could correlate the interannual variability of streamflow statistics to major atmosphere/ocean indices. I'm not familiar with which ones are important for Australia, but ones that are known or suspected to be important to rainfall or streamflows could be tested. These could be relatively easy and may provide valuable information for interpreting the step changes. The discussion could also focus on the timing of known changes (what year) for indices that are important to Australian hydrology and compare those to the years that catchments showed step changes.*

**Answer**

This is already addressed above in general comments, more discussion were added for this point.

**Changes:**

- Literatures on climate indices/changes added in References and discussion added in Section 5.2

*Line 396, It seems like the text describing trends for different regions doesn't match Figure 5. Rather than "Northern Territory and north-west of Western Australia, shouldn't it be "northern part of Northern Territory"? There's only one weak trend in northern Western Australia.*

**Answer**

We have modified the text to be more specifically describing the locations.

**Changes:**

- Text revised in Section 6, line 521-531 (track-change version)

*Line 401, Catchments in the southeast of S. Australia have significant downward trends in Figure 5.*

**Answer**

We have included it in the text.

**Changes:**
• Text revised in Section 6, line 521-531 (track-change version)
*Line 413, Both areas have a mix of step changes in the 1990s and 1970s in Figure 6.*
*General comment on figures: the trend symbols are too small in Figures 5-8.*
*Technical corrections and typos*
**Answer**
We have added this comment in the paragraph, and improve the figure quality (enlarge the
symbol, size of graph, and improve the resolution).
**Changes:**
• Text revised in Section 6, line 543-548 (track-change version)
• Figure 5-9, picture quality improved, with larger size of symbols
*Line 396, incorrect figure reference.*
**Answer**
It's been corrected as "Figure 1". Thanks for pointing it out.
**Changes:**
• corrected in Section 6, line 521 (track-change version)
*Figure 5 caption, change "decrease" to "decreasing"*
**Answer**
We have changed the caption accordingly.
**Changes:**
• Corrected in Figure 5 caption
Thank you again for your valuable comments!

**2. Authors' response and changes to Referee #2**

(Anonymous Referee #2,; Editor's decision to compile all the changes and upload the revised manuscript, received 30 June 2016)

**General comments**

The overall impression of this paper is that it is very clear, well-structured and interesting. The topic of temporal hydrologic change is highly relevant, and the quantitative data analysis of 222 stream gauges is comprehensive and previously unprecedented.

The paper presents a neat compilation of a large quantity of data and addresses relevant scientific questions within the scope of HESS – both regarding the issues of temporal hydrologic change, but also the central question regarding aggregation, compilation and presentation of large data quantities (daily discharge values for 222 stations for 45 years).

The presentation of the HRS web portal great! This is a valuable resource, which will be of great use for the international hydrological community. A paper such as "How streamflow has changed across Australia::" will (apart from its research significance in other ways) have an additional value of helping more researches find the publicly available Australian discharge data.

The paper is written in a clear, concise and straightforward manner, answering most questions that arise. The title clearly reflects the contents of the paper. The language is (as far as I can judge) fluent and correct, the paper is generally very readable. The mathematical formulae, symbols, abbreviations, and units correctly are correctly defined and used. The length of the paper is exemplary short, but still comprehensive enough.

The abstract provides a concise and complete summary, although I'm slightly confused about the expression 'living gauges'.

The scientific methods and assumptions are valid and clearly outlined, allowing reproduction (and traceability of results, as all data and used equations are publicly accessible).

The statistical methods are thoroughly explained, and the decision to have these equations in an appendix is wise. The amount and quality of supplementary material is considered appropriate, and the figures and tables are generally in good shape, and are referred to accordingly.

In general, the number and quality of references seems appropriate for the topic, even though I think that a few more references regarding climate change could have been provided. Especially, I miss a reference to the most recent IPCC which would be of value here.

The scientific approach and the applied methods are valid and the results are to be sufficient to support the interpretations, and the substantial conclusions that are reached.

**Answer**

The authors would like to thank the referee for those positive evaluations of the manuscript and our work; and for the insightful comments on the data and method.

For the first question in general comments about 'living gauges': we have avoided using the expression 'living gauges', and specify its meaning more clearly to avoid confusion (it was changed to "critically important gauges", at line 29 and line 93).

For the second question in general comments, we have added a few more references regarding climate change, including:

(1) CSIRO and Bureau of Meteorology (2015) *Climate Change in Australia Information for Australia's Natural Resource Management Regions: Technical Report.* CSIRO and Bureau of Meteorology, Australia 222pp. http://www.climatechangeinaustralia.gov.au/

(2) Bureau of Meteorology (2016) Annual Climate Report 2015
http://www.bom.gov.au/climate/annual_sum/2015/Annual-Climate-Report-2015-LR.pdf

**Changes:**
- expression 'living gauges' was removed, text revised in Abstract, line 30; and in Section 1, line 96 (track-change version)
- a recent IPCC report added in References and in text

**Specific comments**

My primary concern regards the limited reasoning regarding how the temporal change in streamflow is interrelated to a temporal change in precipitation.
The authors mention clearly that this is not within the scope of the study – which of course is fine. However, the dry period in the last decade in the south-eastern and south-western region is mentioned as a cause of some of the general downward trend.
Although a thorough analysis is of course not viable within the scope of this paper, it would be nice to (if possible) have some discussion regarding the likeliness of this downward trend only being a consequence of the rainfall during a few dry years, or if the trend is likely to be consistent in the longer time perspective. Looking at table 2, at the years of the step change – 1996 is clearly the most dominating year (13 of 22!): an added reflection regarding the impacts of this (probably very non-normal) hydrological year would be interesting. How much impact does this "outlier year" have on the temporal trend? Would the same general pattern be seen even if it was to be omitted from the analysis? I do not request you to do the complete analysis of this issue, but some kind of (further) discussion on the topic could be useful.

**Answer**
Many thanks for the suggestion. We agree that this ought to be discussed. Though a thorough analysis is out of scope of this paper, we have added relevant literatures on climate (as mentioned above), and included discussion on relating the flow changes with rainfall, and more discussion on the finding of large numbers of 1996 step changes in southeast Australia linking to the millennium drought (SEACI, 2011, The Millennium Drought and 2010/11 Floods, http://www.seaci.org/publications/documents/SEACI-2Reports/SEACI2_Factsheet2of4_WEB_110714.pdf).

**Changes:**
- discussion added on linking flow and rain trend, see Section 5.2
- more literature on climate added

Also, I believe that most data is available from the 1950's and onwards. However, I guess that longer time series should be available at least for some gauges. A comparison regarding an even more long-term time series would give additional weight to the results – although, this may be the subject of another study.

**Answer**
Yes, there are some stations which have longer time series but not many. More long-term time series will certainly give additional weight to the results, as you said, it will be an interesting point to be added for future study.

Line 152 – please also add the median time-series length.

**Answer**
The median time-series length (46.6 years) was added in this part.

**Changes:**
- median time-series length added in Section 2.1, line 168

Lines 206-208 – is any of this presented here? Or mainly as background info to the tables/figures?

**Answer**
These statistical data analyses in lines 204-208 were only mentioned here as background
information for all types of graphic products in HRS web portal. For more details of other
statistical data analyses that's not presented in this study, please check the information at
the HRS website: http://www.bom.gov.au/water/hrs/
Line 262 – shouldn't also land-use changes be mentioned in this context?
**Answer**
Thanks for pointing it out. We have added discussion on land use changes in this context.
**Changes:**
• text added in Section 4.2, line 300-301
One last comment: the fact that different hydrologic years are used for different stations (if I
understand it correct) – will this have an impact on the results (lines 149-151)?
**Answer**
A quick answer to this is "No, not much impact on the results".
Water year or hydrologic year was used in this study, but it's not different for every station.
Table 1 has listed the water year start month for each division, and they are in a more
consistent way: for regions in the south part of Australia, water year starts at March or
February; for regions in the north and central Australia, it starts at September or October. In
this way, the analyses were more following the natural hydrologic pattern, and representing
the results in a better way.
**Technical corrections**
There are hardly any technical corrections that need to be addressed in the paper. The
authors have made a robust study, and compiled the data in a presentable and concise
manner.
I am however not clear about what the authors mean by the concept of 'living gauges',
neither in the abstract nor in the text (lines 29 and 93) – don't just normal gauges record and
detect changes in hydrologic responses?
**Answer**
This has been addressed in the general comments above.
As not being very familiar with Australian geography, I would have appreciated (if possible to
do in an aesthetic manner) information regarding the names of the basins in figure 1 –
perhaps by inserting the roman numerals from table 1 on the map?
**Answer**
We have modified Figure 1 in this way, by inserting the basin code for each region, and
readers can refer the basin names to Table1.
**Changes:**
• Figure 1 was modified with drainage division basin names (green colour font
matching the colour of drainage division borders) labelled on the map
• Figure 5 was modified with Australian state names (grey colour font matching the
colour of Australian state borders) labelled on the map
Also, table 2 seems to be of somewhat low resolution (the letters are blurry) – if possible,
please improve this.
**Answer**
We have improved the quality of Table 2. Table 2 in the submitted manuscript was actually a
graph (that's why the letters look blurry), as we had difficulties to inset the text table in a
landscape layout. We have updated it in the revised version.
**Changes:**

•    Table 2, content updated and quality improved. We will upload the original Word file
of Table 2 as a separate file
Figure 5 (and 6 and 8), please add Q_(appropriate index) in the text for clarity.
Thanks for a good read, and congratulations on your thorough study! I'm looking forward to
seeing more of this paper in the future!
**Answer**
We have added Q_(appropriate index) in the figures, for readers could easily refer to.
**Changes:**
•    Figure 5,6,8,9, Q_(appropriate index) was specified in figures and figure captions

**3. Authors' response and changes to Short comments from M. Hipsey**

(Short comments from M. Hipsey, matt.hipsey@uwa.edu.au;; Editor's decision to compile all the changes and upload the revised manuscript, received 30 June 2016)

Thank you for your time to review our manuscript. You have mentioned valuable points, which we really appreciate. Please find our response below to your comments, questions and suggestions. The referee's comments are first recalled in *italics, blue colour font*, and then followed by our **answer** and **changes**.

*As an Australian, I have read the submission with great interest and was pleased to see the analysis undertaken. I highlight the substantial amount of work that has gone into curating and making sense of such a large dataset at this scale. This is important progress and important not only for scientific purposes but for shaping policy in Australia.*
*I would like to make a few short suggestions that could be considered during the discussion/revision process.*
*I think there are some problems with the section headings. Aside from the fact the sub-sub-heading is larger font than the sub-heading; I also note that section 4 is*
*"Results and Discussion" and section 5 is "Discussion" ... There is also a Section 6 with "Conclusions". I suggest these 3 sections and their sub-headings could be carefully looked at, and would suggest splitting results and discussion into separate sections, with sub-headings used in the discussion to help navigate the reader to the significant findings.*

**Answer**
Thank you for noticing this and helpful suggestions to improve the structure of manuscript. We will adjust all the headings at different levels in a systematic way to reflect the hierarchy structure clearly. Also the section or sub-section titles will be modified.

**Changes:**
- All headings format adjusted
- Sections and sub-sections re-arranged, with Section 4 – Results, Section 5 – Discussion, Section 6 – Conclusion, and sub-sections to help navigate the reader to different discussion points

*Further from the above, the aim as stated at the end of the introduction is to provide a nationwide assessment of trends in streamflow which is achieved well. Of course one of the powers of compiling the dataset is to the try tease out the science of why trends are occurring and it would be nice to see this as an aim. I notice a brief paragraph on this point (Page 13) highlighting general drying trend in the climate etc, but I felt the study would become much more powerful if there was a more significant attempt to explain the non-stationary behaviour. This could range from a quantitative assessment of changes in the rainfall-runoff coefficient (is the streamflow change amplifying or dampening the broad rainfall trends in each region?) or at a minimum could consist of a more detailed and focused discussion on Page 13 introducing and citing previous studies explaining mechanisms for the trends. For example, Smettem et al 2013 undertook an analysis on the forest response to drying trend impacts streamflow; Ummenhofer et al., 2009 on mechanisms for increasing drought; there are obviously many more papers relevant to different regions that could help readers understand the mechanism and significance of the trend. It is stated as being beyond the scope (in ln376), however, I would suspect it would be of key interest to the HESS readership and I would suggest that space could be made by moving section 3 and Figure 3 to an Appendix; in fact I would encourage the authors to refocus the aims on the hydrological trends AND their explanations, rather than the focus on the web portal itself.*

**Answer**

We agree to this point. Though a thorough investigation of reasons behind the hydrological trends is beyond scope of this article, we added relevant literatures on past climate changes, non-stationarity in streamflow Australia (including the papers you mentioned - Smettem et al 2013; Ummenhofer et al., 2009), and extend the discussion accordingly, also to relate the flow changes with rainfall.

About section 3 on web portal development, it provides important information for key users of this study. It will be kept in the main text however we will make it concise.

**Changes:**

- discussion added on linking flow and rain trend, see Section 5.2
- literatures added in References, including the above mentioned ones

Lastly, whilst it is related to the above, it would be ideal for the discussion to cover the projections of climate change for the different regions to address the question of whether the past changes are likely to continue, and as justification for the ongoing monitoring and assessment at the nation-wide scale. This need not be an extensive addition, just some targeted references cited for interested readers, potentially within a dedicated sub-section in the discussion.

Thank you very much for the opportunity to comment on this great study, and I do hope these comments will be seen as constructive criticisms to help improve the overall paper and usefulness of the analysis.

**Answer**

This study is focused on the flow changes that we observed from the historical data records, and not trying to refer it to the future. From the sustainable yield study done by CSIRO (http://www.csiro.au/en/Research/LWF/Areas/Water-resources/Assessing-water-resources/Sustainable-yields ), It is likely that rainfall/streamflow trends in  the Murray-Darling Basin, southern Australia and south-west Western Australia is likely to continue.

The suggestion you raised here will be an interesting point to look at, but unfortunately it's out of scope of this paper. We added relevant literatures on past climate changes (as mentioned above), and included discussion on relating the flow changes with rainfall trend, which we hope will provide some useful information for readers to understand and interpret the trends and step changes presented.

Thanks again for your valuable comments!

Here below attached a marked-up manuscript with track changes.

[revised manuscript text omitted]

Q_BF: annual baseflow, Qmax: daily maximum flow, Q90: 90th percentile daily flow, Q50:

50th percentile daily flow,Q10: 10th percentile daily flow, Q_DJF: summer flow, Q_MAM:

autumn flow, Q_JJA: winter flow, Q_SON: spring flow)

[Figure]

[Figure]

Figure 5: Spatial variation in trend results of annual total flow(Q_T), decreasinge trends were shown in significance levels at 0.01, 0.05, and 0.1

[Figure]

[Figure]

Figure 6: Variations of step change in annual total flow $(Q_T)$ for stations showing significant increase or decrease trend

[Figure]

Figure 7: a) Percentage and b) number of stations showing significant upward and downward trends or step changes in different regions.

[Figure]

a) Trend of maximum daily flow Qmax
⊛ Non-random series
• No significant trend
▼ Decreasing trend
▲ Increasing trend b) Trend of Q90 daily flow
⊛ Non-random series
• No significant trend
▼ Decreasing trend
▲ Increasing trend

c) Trend of Q50 daily flow
⊛ Non-random series
• No significant trend
▼ Decreasing trend
▲ Increasing trend d) Trend of Q10 daily flow
⊛ Non-random series
• No significant trend
▼ Decreasing trend
▲ Increasing trend

a) Trend of maximum daily flow Qmax
⊛ Non-random series
• No significant trend
▼ Decreasing trend
▲ Increasing trend b) Trend of Q90 daily flow
⊛ Non-random series
• No significant trend
▼ Decreasing trend
▲ Increasing trend

[Figure]

Figure 87: Maps showing trends of daily flow in various magnitude categories a) maximum daily flow $Q_{Max}$; b) $Q_{90}$ daily flow; c) $Q_{50}$ daily flow; d) $Q_{10}$ daily flow at 10% significant level (p<0.1)

[Figure]

[Figure]

Figure 98: Maps showing trends of seasonal flow in a) Summer (QDJF); b) Autumn (QMAM); c) Winter (QJJA); d) Spring (QSON)